# Learning Efficient Online 3D Bin Packing on Packing Configuration Trees

**Hang Zhao**[1,2*]**, Yang Yu**[2]**, Kai Xu**[1†]
[1]School of Computer Science, National University of Defense Technology, China
[2]National Key Lab for Novel Software Technology, Nanjing University, China
{alex.hang.zhao, kevin.kai.xu}@gmail.com, yuy@nju.edu.cn

## Abstract

Online 3D Bin Packing Problem (3D-BPP) has widespread applications in industrial automation and has aroused enthusiastic research interest recently. Existing methods usually solve the problem with limited resolution of spatial discretization, and/or cannot deal with complex practical constraints well. We propose to enhance the practical applicability of online 3D-BPP via learning on a novel hierarchical representation — packing configuration tree (PCT). PCT is a full-fledged description of the state and action space of bin packing which can support packing policy learning based on deep reinforcement learning (DRL). The size of the packing action space is proportional to the number of leaf nodes, i.e. candidate placements, making the DRL model easy to train and well-performing even with continuous solution space. During training, PCT expands based on heuristic rules, however, the DRL model learns a much more effective and robust packing policy than heuristic methods. Through extensive evaluation, we demonstrate that our method outperforms all existing online BPP methods and is versatile in terms of incorporating various practical constraints.

## 1 Introduction

As one of the most classic combinatorial optimization problems, the 3D bin packing problem usually refers to packing a set of cuboid-shaped items $i \in \mathcal{I}$, with sizes $s_i^x, s_i^y, s_i^z$ along $x, y, z$ axes, respectively, into the minimum number of bins with sizes $S^x, S^y, S^z$, in an axis-aligned fashion. Traditional 3D-BPP assumes that all the items to be packed are known a priori (Martello et al., 2000), which is also called *offline* BPP. The problem is known to be strongly NP-hard (De Castro Silva et al., 2003). However, in many real-world application scenarios, e.g., logistics or warehousing (Wang & Hauser, 2019a), the upcoming items cannot be fully observed; only the current item to be packed is observable. Packing items without the knowledge of all upcoming items is referred to as *online* BPP (Seiden, 2002).

Due to its obvious practical usefulness, online 3D-BPP has received increasing attention recently. Given the limited knowledge, the problem cannot be solved by usual search-based methods. Different from offline 3D-BPP where the items can be placed in an arbitrary order, online BPP must place items following their coming order, which imposes additional constraints. Online 3D-BPP is usually solved with either heuristic methods (Ha et al., 2017) or learning-based ones (Zhao et al., 2021), with complementary pros and cons. Heuristic methods are generally not limited by the size of action space, but they find difficulties in handling complex practical constraints such as packing stability or specific packing preferences. Learning-based approaches usually perform better than heuristic methods, especially under various complicated constraints. However, the learning is hard to converge with a large action space, which has greatly limited the applicability of learning-based methods due to, e.g., the limited resolution of spatial discretization (Zhao et al., 2021).

We propose to enhance learning-based online 3D-BPP towards practical applicability through learning with a novel hierarchical representation — *packing configuration tree (PCT)*. PCT is a dynamically growing tree where the internal nodes describe the space configurations of packed items and

---

*Work conducted while the author was visiting the National Key Lab for Novel Software Technology.
†Kai Xu is the corresponding author.

leaf nodes the packable placements of the current item. PCT is a full-fledged description of the state and action space of bin packing which can support packing policy learning based on deep reinforcement learning (DRL). We extract state features from PCT using graph attention networks (Velickovic et al., 2018) which encodes the spatial relations of all space configuration nodes. The state feature is input into the actor and critic networks of the DRL model. The actor network, designed based on pointer mechanism, weighs the leaf nodes and outputs the action (the final placement).

During training, PCT grows under the guidance of heuristics such as *Corner Point* (Martello et al., 2000), *Extreme Point* (Crainic et al., 2008), and *Empty Maximal Space* (Ha et al., 2017). Although PCT is expanded with heuristic rules, confining the solution space to what the heuristics could explore, our DRL model learns a discriminant fitness function (the actor network) for the candidate placements, resulting in an effective and robust packing policy exceeding the heuristic methods. Furthermore, the size of the packing action space is proportional to the number of leaf nodes, making the DRL model easy to train and well-performing even with continuous solution space where the packing coordinates are continuous values. Through extensive evaluation, we demonstrate that our method outperforms all existing online 3D-BPP methods and is versatile in terms of incorporating various practical constraints such as isle friendliness and load balancing (Gzara et al., 2020). Our work is, to our knowledge, the first that deploys the learning-based method on solving online 3D-BPP with continuous solution space successfully.

## 2 RELATED WORK

**Offline 3D-BPP**   The early interest of 3D-BPP mainly focused on its offline setting. Offline 3D-BPP assumes that all items are known as a priori and can be placed in an arbitrary order. Martello et al. (2000) first solved this problem with an exact branch-and-bound approach. Limited by exponential worst-case complexity of exact approaches, lots of heuristic and meta-heuristic algorithms are proposed to get an approximate solution quickly, such as guided local search (Faroe et al., 2003), tabu search (Crainic et al., 2009), and hybrid genetic algorithm (Kang et al., 2012). Hu et al. (2017) decompose the offline 3D-BPP into packing order decisions and online placement decisions. The packing order is optimized with an end-to-end DRL agent and the online placement policy is a hand-designed heuristic. This two-step fashion is widely accepted and followed by Duan et al. (2019), Hu et al. (2020), and Zhang et al. (2021).

**Heuristics for Online 3D-BPP**   Although offline 3D-BPP has been well studied, their search-based approaches cannot be directly transferred to the online setting. Instead, lots of heuristic methods have been proposed to solve this problem. For reasons of simplicity and good performance, the deep-bottom-left (DBL) heuristic (Karabulut & Inceoglu, 2004) has long been a favorite. Ha et al. (2017) sort the empty spaces with this DBL order and place the current item into the first fit one. Wang & Hauser (2019b) propose a Heightmap-Minimization method to minimize the volume increase of the packed items as observed from the loading direction. Hu et al. (2020) optimize the empty spaces available for the packing future with a Maximize-Accessible-Convex-Space method.

**DRL for Online 3D-BPP**   The heuristic methods are intuitive to implement and can be easily applied to various scenarios. However, the price of good flexibility is that these methods perform mediocrely, especially for online 3D-BPP with specific constraints. Designing new heuristics for specific classes of 3D-BPP is heavy work since this problem has an NP-hard solution space, many situations need to be premeditated manually by trial and error. Substantial domain knowledge is also necessary to ensure safety and reliability. To automatically generate a policy that works well on specified online 3D-BPP, Verma et al. (2020); Zhao et al. (2021) employ the DRL method on solving this problem, however, their methods only work in small discrete coordinate spaces. Despite their limitations, these works are soon followed by Hong et al. (2020); Yang et al. (2021); Zhao et al. (2022) for logistics robot implementation. Zhang et al. (2021) adopt a similar online placement policy for offline packing needs referring to Hu et al. (2017). All these learning-based methods only work in a grid world with limited discretization accuracy, which reduces their practical applicability.

**Practical Constraints**   The majority of literature for 3D-BPP (Martello et al., 2000) only considers the basic non-overlapping constraint 1 and containment constraint 2:

$$p_i^d + s_i^d \le p_j^d + S^d(1 - e_{ij}^d) \quad i \ne j, i, j \in \mathcal{I}, d \in \{x, y, z\} \tag{1}$$

$$0 \le p_i^d \le S^d - s_i^d \quad i \in \mathcal{I}, d \in \{x, y, z\} \tag{2}$$

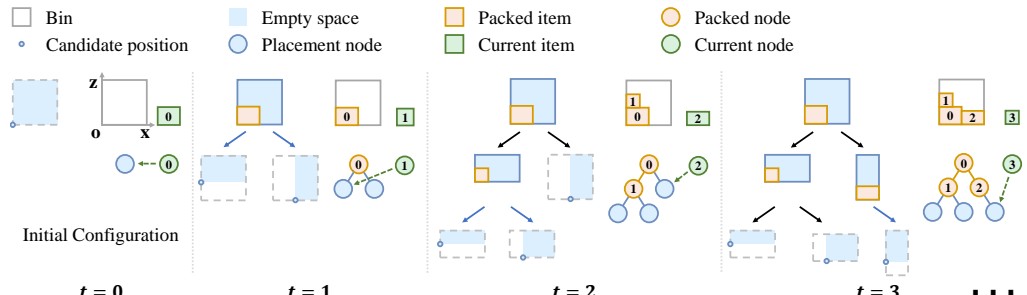

Figure 1: PCT expansion illustrated using a 2D example (in $xoz$ plane) for simplicity and the number of allowed orientations $|\mathbf{O}|$ is 1 (see Appendix B for the 3D version). A newly added item introduces a series of empty spaces and new candidate placements are generated, e.g., the left-bottom corner of the empty space.

Where $p_i$ means the front-left-bottom coordinate of item $i$ and $d$ the coordinate axis, $e_{ij}$ takes value 1 otherwise 0 if item $i$ precedes item $j$ along $d$. The algorithms for 3D-BPP are of limited practical applicability if no even basic real-world constraints, e.g., stability (Ramos et al., 2016), are considered. Zhao et al. (2022) propose a fast stability estimation method for DRL training and test their learned policies with real logistics boxes. The flaw of their work is the heightmap (the upper frontier of packed items) state representation like Zhang et al. (2021) is still used, while the underlying constraints between packed items are missed. The unavailability of underlying spatial information makes their problem a partially observable Markov Decision Process (Spaan, 2012) which is not conducive to DRL training and limits the performance on 3D-BPP instances with more complex practical constraints, like isle friendliness and load balancing (Gzara et al., 2020).

## 3    METHOD

In this section, we first introduce our PCT concept in Section 3.1 for describing the online packing process. The parameterization of the tree structure and the leaf node selection policy are introduced in Section 3.2 and Section 3.3 respectively. In Section 3.4, we formulate online 3D-BPP as Markov Decision Process based on PCT, followed by the description of the training method.

### 3.1    PACKING CONFIGURATION TREE

When a rectangular item $n_t$ is added to a given packing with position $(p_n^x, p_n^y, p_n^z)$ at time step $t$, it introduces a series of new candidate positions where future items can be accommodated, as illustrated in Figure 1. Combined with the axis-aligned orientation $o \in \mathbf{O}$ for $n_t$ based on existing positions, we get candidate placements (i.e. position and orientation). The packing process can be seen as a placement node being replaced by a packed item node, and new candidate placement nodes are generated as children. As the packing time step $t$ goes on, these nodes are iteratively updated and a dynamic *packing configuration tree* is formed, denoted as $\mathcal{T}$. The internal node set $\mathbf{B}_t \in \mathcal{T}_t$ represents the space configurations of packed items, and the leaf node set $\mathbf{L}_t \in \mathcal{T}_t$ the packable candidate placements. During the packing, leaf nodes that are no longer feasible, e.g., covered by packed items, will be removed from $\mathbf{L}_t$. When there is no packable leaf node that makes $n_t$ satisfy the constraints of placement, the packing episode ends. Without loss of generality, we stipulate a vertical top-down packing within a single bin (Wang & Hauser, 2019b).

Traditional 3D-BPP literature only cares about the remaining placements for accommodating the current item $n_t$, their packing policies can be written as $\pi(\mathbf{L}_t | \mathbf{L}_t, n_t)$. If we want to promote this problem for practical demands, 3D-BPP needs to satisfy more complex practical constraints which also act on $\mathbf{B}_t$. Taking packing stability for instance, a newly added item $n_t$ has possibly force and torque effect on the whole item set $\mathbf{B}_t$ (Ramos et al., 2016). The addition of $n_t$ should make $\mathbf{B}_t$ a more stable spatial distribution so that more items can be added in the future. Therefore, our packing policy over $\mathbf{L}_t$ is defined as $\pi(\mathbf{L}_t | \mathcal{T}_t, n_t)$, which means probabilities of selecting leaf nodes from $\mathbf{L}_t$ given $\mathcal{T}_t$ and $n_t$. For online packing, we hope to find the best leaf node selection policy to expand the PCT with more relaxed constraints so that more future items can be appended.

**Leaf Node Expansion Schemes**    The performance of online 3D-BPP policies has a strong relationship with the choice of leaf node expansion schemes — which incrementally calculate new candidate placements introduced by the just placed item $n_t$. A good expansion scheme should reduce

the number of solutions to be explored while not missing too many feasible packings. Meanwhile, polynomially computability is also expected. Designing such a scheme from scratch is non-trivial. Fortunately, several placement rules independent from particular packing problems have been proposed, such as *Corner Point* (Martello et al., 2000), *Extreme Point* (Crainic et al., 2008), and *Empty Maximal Space* (Ha et al., 2017). We extend these schemes which have proven to be accurate and efficient to our PCT expansion. The performance of learned policies will be reported in Section 4.1.

## 3.2 TREE REPRESENTATION

Given the bin configuration $\mathcal{T}_t$ and the current item $n_t$, the packing policy can be parameterized as $\pi(\mathbf{L}_t|\mathcal{T}_t, n_t)$. The tuple $(\mathcal{T}_t, n_t)$ can be treated as a graph and encoded by Graph Neural Networks (GNNs) (Gori et al., 2005). Specifically, the PCT keeps growing with time step $t$ and cannot be embedded by spectral-based approaches (Bruna et al., 2014) which require a fixed graph structure. We adopt non-spectral Graph Attention Networks (GATs) (Velickovic et al., 2018), which require no priori on graph structures.

The raw space configuration nodes $\mathbf{B}_t, \mathbf{L}_t, n_t$ are presented by descriptors in different formats. We use three independent node-wise Multi-Layer Perceptron (MLP) blocks to project these heterogeneous descriptors into the homogeneous node features: $\hat{\mathbf{h}} = \{\phi_{\theta_B}(\mathbf{B}_t), \phi_{\theta_L}(\mathbf{L}_t), \phi_{\theta_n}(n_t)\} \in \mathbb{R}^{d_h \times N}$, $d_h$ is the dimension of each node feature and $\phi_\theta$ is an MLP block with its parameters $\theta$. The feature number $N$ should be $|\mathbf{B}_t| + |\mathbf{L}_t| + 1$, which is a variable. The GAT layer is used to transform $\hat{\mathbf{h}}$ into high-level node features. The Scaled Dot-Product Attention (Vaswani et al., 2017) is applied to each node for calculating the relation weight of one node to another. These relation weights are normalized and used to compute the linear combination of features $\hat{\mathbf{h}}$. The feature of node $i$ embedded by the GAT layer can be represented as:

$$\text{GAT}(\hat{h}_i) = W^O \sum_{j=1}^N softmax\left(\frac{(W^Q\hat{h}_i)^T W^K \hat{h}_j}{\sqrt{d_k}}\right) W^V \hat{h}_j \tag{3}$$

Where $W^Q \in \mathbb{R}^{d_k \times d_h}$, $W^K \in \mathbb{R}^{d_k \times d_h}$, $W^V \in \mathbb{R}^{d_v \times d_h}$, and $W^O \in \mathbb{R}^{d_h \times d_v}$ are projection matrices, $d_k$ and $d_v$ are dimensions of projected features. The softmax operation normalizes the relation weight between node $i$ and node $j$. The initial feature $\hat{\mathbf{h}}$ is embedded by a GAT layer and the skip-connection operation (Vaswani et al., 2017) is followed to get the final output features $\mathbf{h}$:

$$\mathbf{h}' = \hat{\mathbf{h}} + \text{GAT}(\hat{\mathbf{h}}) \quad \mathbf{h} = \mathbf{h}' + \phi_{FF}(\mathbf{h}') \tag{4}$$

Where $\phi_{FF}$ is a node-wise Feed-Forward MLP with output dimension $d_h$ and $\mathbf{h}'$ is an intermediate variable. Equation 4 can be seen as an independent block and be repeated multiple times with different parameters. We don't extend GAT to employ the multi-head attention mechanism (Vaswani et al., 2017) since we find that additional attention heads cannot help the final performance. We execute Equation 4 once and we set $d_v = d_k$. More implementation details are provided in Appendix A.

## 3.3 LEAF NODE SELECTION

Given the node features $\mathbf{h}$, we need to decide the leaf node indices for accommodating the current item $n_t$. Since the leaf nodes vary as the PCT keeps growing over time step $t$, we use a pointer mechanism (Vinyals et al., 2015) which is context-based attention over variable inputs to select a leaf node from $\mathbf{L}_t$. We still adopt Scaled Dot-Product Attention for calculating pointers, the global context feature $\bar{h}$ is aggregated by a mean operation on $\mathbf{h}$: $\bar{h} = \frac{1}{N}\sum_{i=1}^N h_i$. The global feature $\bar{h}$ is projected to a query $q$ by matrix $W^q \in \mathbb{R}^{d_k \times d_h}$ and the leaf node features $\mathbf{h_L}$ are utilized to calculate a set of keys $k_\mathbf{L}$ by $W^k \in \mathbb{R}^{d_k \times d_h}$. The compatibility $\mathbf{u_L}$ of the query with all keys are:

$$q = W^q\bar{h} \quad k_i = W^k h_i \quad u_i = \frac{q^T k_i}{\sqrt{d_k}} \tag{5}$$

Here $h_i$ only comes from $\mathbf{h_L}$. The compatibility vector $\mathbf{u_L} \in \mathbb{R}^{|\mathbf{L}_t|}$ represents the leaf node selection logits. The probability distribution over the PCT leaf nodes $\mathbf{L}_t$ is:

$$\pi_\theta(\mathbf{L}_t|\mathcal{T}_t, n_t) = softmax\left(c_{clip} \cdot \tanh\left(u_\mathbf{L}\right)\right) \tag{6}$$

Following Bello et al. (2017), the compatibility logits are clipped with tanh, where the range is controlled by hyperparameter $c_{clip}$, and finally normalized by a softmax operation.

### 3.4 MARKOV DECISION PROCESS FORMULATION

The online 3D-BPP decision at time step $t$ only depends on the tuple $(\mathcal{T}_t, n_t)$ and can be formulated as Markov Decision Process, which is constructed with state $\mathcal{S}$, action $\mathcal{A}$, transition $\mathcal{P}$, and reward $R$. We solve this MDP with an end-to-end DRL agent. The MDP model is formulated as follows.

**State**  The state $s_t$ at time step $t$ is represented as $s_t = (\mathcal{T}_t, n_t)$, where $\mathcal{T}_t$ consists of the internal nodes $\mathbf{B}_t$ and the leaf nodes $\mathbf{L}_t$. Each internal node $b \in \mathbf{B}_t$ is a spatial configuration of sizes $(s_b^x, s_b^y, s_b^z)$ and coordinates $(p_b^x, p_b^y, p_b^z)$ corresponding to a packed item. The current item $n_t$ is a size tuple $(s_n^x, s_n^y, s_n^z)$. Extra properties will be appended to $b$ and $n_t$ for specific packing preferences, such as density, item category, etc. The descriptor for leaf node $l \in \mathbf{L}_t$ is a placement vector of sizes $(s_o^x, s_o^y, s_o^z)$ and position coordinates $(p^x, p^y, p^z)$, where $(s_o^x, s_o^y, s_o^z)$ indicates the sizes of $n_t$ along each dimension after an axis-aligned orientation $o \in \mathbf{O}$. Only the packable leaf nodes which satisfy placement constraints are provided.

**Action**  The action $a_t \in \mathcal{A}$ is the index of the selected leaf node $l$, denoted as $a_t = index(l)$. The action space $\mathcal{A}$ has the same size as $\mathbf{L}_t$. A surge of learning-based methods (Zhao et al., 2021) directly learn their policy on a grid world through discretizing the full coordinate space, where $|\mathcal{A}|$ grows explosively with the accuracy of the discretization. Different from existing works, our action space solely depends on the leaf node expansion scheme and the packed items $\mathbf{B}_t$. Therefore, our method can be used to solve online 3D-BPP with continuous solution space. We also find that even if only an intercepted subset $\mathbf{L}_{sub} \in \mathbf{L}_t$ is provided, our method can still maintain a good performance.

**Transition**  The transition $\mathcal{P}(s_{t+1}|s_t)$ is jointly determined by the current policy $\pi$ and the probability distribution of sampling items. Our online sequences are generated on-the-fly from an item set $\mathcal{I}$ in a uniform distribution. The generalization performance of our method on item sampling distributions different from the training one is discussed in Section 4.4.

**Reward**  Our reward function $R$ is defined as $r_t = c_r \cdot w_t$ once $n_t$ is inserted into PCT as an internal node successfully, otherwise, $r_t = 0$ and the packing episode ends. Here $c_r$ is a constant and $w_t$ is the weight of $n_t$. The choice of $w_t$ depends on the customized needs. For simplicity and clarity, unless otherwise noted, we set $w_t$ as the volume occupancy $v_t$ of $n_t$, where $v_t = s_n^x \cdot s_n^y \cdot s_n^z$.

**Training Method**  A DRL agent seeks for a policy $\pi(a_t|s_t)$ to maximize the accumulated discounted reward. Our DRL agent is trained with the ACKTR method (Wu et al., 2017). The actor weighs the leaf nodes $\mathbf{L}_t$ and outputs the policy distribution $\pi_\theta(\mathbf{L}_t|\mathcal{T}_t, n_t)$. The critic maps the global context $\bar{h}$ into a state value prediction to predict how much accumulated discount reward the agent can get from $t$ and helps the training of the actor. The action $a_t$ is sampled from the distribution $\pi_\theta(\mathbf{L}_t|\mathcal{T}_t, n_t)$ for training and we take the argmax of the policy for the test.

ACKTR runs multiple parallel processes for gathering on-policy training samples. The node number $N$ of each sample varies with the time step $t$ and the packing sequence of each process. For batch calculation, we fullfill PCT to a fixed length with dummy nodes, as illustrated by Figure 2 (a). These redundant nodes are eliminated by *masked attention* (Velickovic et al., 2018) during the feature calculation of GAT. The aggregation of $\mathbf{h}$ only happens on the eligible nodes. For preserving node spatial relations, state $s_t$ is embedded by GAT as a fully connected graph as Figure 2 (b), without any inner mask operation. More implementation details are provided in Appendix A.

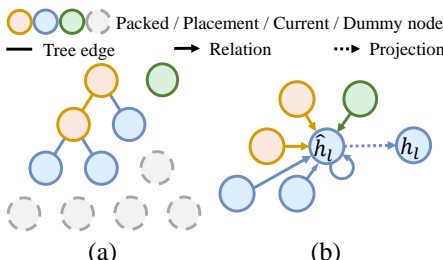

Figure 2: Batch calculation for PCT.

## 4 EXPERIMENTS

In this section, we first report the performance of our PCT model combined with different leaf node expansion schemes. Then we will demonstrate the benefits of the PCT structure: better node spatial relation representations as well as more flexible action space. Finally, we test the generalization performance of our method and extend it to other online 3D-BPP with complex practical constraints.

**Baselines**  Although there are very few online packing implementations publicly available, we still do our best to collect or reproduce various online 3D-BPP algorithms, both heuristic and learning-

based, from potentially relevant literature. We help the heuristic methods to make a pre-judgment of placement constraints, e.g., stability, in case of premature downtime. The learning-based agents are trained until there are no significant performance gains. Although both Zhao et al. (2022) and Zhao et al. (2021) are learning-based methods recently proposed, we only compare with the former since it is the upgrade of the latter. We also compare with the reported results of Zhang et al. (2021) under the same condition. All methods are implemented in Python and tested on 2000 instances with a desktop computer equipped with a Gold 5117 CPU and a GeForce TITAN V GPU. We publish the source code of our method along with related baselines at Github[1].

**Datasets** Some baselines like Karabulut & Inceoglu (2004) need to traverse the entire coordinate space to find the optimal solution, and the running costs explode as the spatial discretization accuracy increases. To ensure that all algorithms are runnable within a reasonable period, we use the discrete dataset proposed by Zhao et al. (2021) without special declaration. The bin sizes $S^d$ are set to 10 with $d \in \{x, y, z\}$ and the item sizes $s^d \in \mathbb{Z}^+$ are not greater than $S^d/2$ to avoid over-simplified scenarios. Our performance on a more complex continuous dataset will be reported in Section 4.3. Considering that there are many variants of 3D-BPP, we choose three representative settings:

$Setting\ 1$: Following Zhao et al. (2022), the stability of the $\mathbf{B}_t$ is checked when $n_t$ is placed. Only two horizontal orientations ($|\mathbf{O}| = 2$) are permitted for robot manipulation convenience.
$Setting\ 2$: Following Martello et al. (2000), item $n_t$ only needs to satisfy Constraint 1 and 2. Arbitrary orientation ($|\mathbf{O}| = 6$) is allowed here. This is the most common setting in 3D-BPP literature.
$Setting\ 3$: Inherited from $setting\ 1$, each item $n_t$ has an additional density property $\rho$ sampled from $(0, 1]$ uniformly. This information is appended into the descriptors of $\mathbf{B}_t$ and $n_t$.

## 4.1 PERFORMANCE OF PCT POLICIES

We first report the performance of our PCT model combined with different leaf node expansion schemes. Three existing schemes which have proven to be both efficient and effective are adopted here: Corner Point (CP), Extreme Point (EP), and Empty Maximal Space (EMS). These schemes are all related to boundary points of a packed item along $d$ axis. We combine the start/end points of $n_t$ with the boundary points of $b \in \mathbf{B}_t$ to get the superset, namely *Event Point* (EV). See Appendix B for details and learning curves. We extend these schemes to our PCT model. Although the number of leaf nodes generated by these schemes is within a reasonable range, we only randomly intercept a subset $\mathbf{L}_{sub_t}$ from $\mathbf{L}_t$ if $|\mathbf{L}_t|$ exceeds a certain length, for saving computing resources. The interception length is a constant during training and determined by a grid search (GS) during the test. See Appendix A for implementation details. The performance comparisons are summarized in Table 1.

Table 1: Performance comparisons. Uti. and Num. mean the average space utilization and the average number of packed items separately. Var. ($\times 10^{-3}$) is the variance of Uti. and Gap is w.r.t. the best Uti. across all methods. *Random* and *Random* & EV randomly pick placements from full coordinates and full EV nodes respectively. DBL, LSAH, MACS, and BR are heuristic methods proposed by Karabulut & Inceoglu (2004), Hu et al. (2017), Hu et al. (2020), and Zhao et al. (2021). Ha et al. (2017), LSAH, and BR are heuristics based on EMS.

| | Method | Setting 1 | | | | Setting 2 | | | | Setting 3 | | | |
|---|---|---|---|---|---|---|---|---|---|---|---|---|---|
| | | Uti. | Var. | Num. | Gap | Uti. | Var. | Num. | Gap | Uti. | Var. | Num. | Gap |
| Heuristic | *Random* | 36.7% | 10.3 | 14.9 | 51.7% | 38.6% | 8.3 | 15.7 | 55.1% | 36.8% | 10.6 | 14.9 | 51.4% |
| | BR | 49.0% | 10.8 | 19.6 | 35.5% | 56.7% | 6.6 | 22.6 | 34.1% | 48.9% | 10.7 | 19.5 | 35.4% |
| | Ha et al. | 52.1% | 20.1 | 20.6 | 31.4% | 59.9% | 10.4 | 23.8 | 30.3% | 51.9% | 20.2 | 20.6 | 31.4% |
| | LSAH | 52.5% | 12.2 | 20.8 | 30.9% | 65.0% | 6.1 | 25.6 | 24.4% | 52.4% | 12.2 | 20.7 | 30.8% |
| | Wang & Hauser | 57.6% | 11.5 | 24.1 | 24.2% | 66.1% | 8.4 | 25.9 | 23.1% | 56.5% | 11.2 | 22.3 | 25.4% |
| | MACS | 57.7% | 10.5 | 22.6 | 24.1% | 50.8% | 8.8 | 20.1 | 40.9% | 57.7% | 10.6 | 22.6 | 23.8% |
| | DBL | 60.5% | 8.8 | 23.8 | 20.4% | 70.6% | 7.9 | 27.8 | 17.9% | 60.5% | 8.9 | 23.8 | 20.1% |
| Learning-based | Zhao et al. | 70.9% | 6.2 | 27.5 | 6.7% | 70.3% | 4.3 | 27.4 | 18.3% | 59.6% | 5.4 | 23.1 | 21.3% |
| | PCT & CP | 69.4% | 5.4 | 26.7 | 8.7% | 81.8% | 2.0 | 31.3 | 4.9% | 69.5% | 5.4 | 26.7 | 8.2% |
| | PCT & EP | 71.9% | 6.6 | 27.8 | 5.4% | 78.1% | 3.8 | 30.3 | 9.2% | 72.2% | 5.8 | 27.9 | 4.6% |
| | PCT & FC | 72.4% | 4.7 | 28.0 | 4.7% | 76.9% | 3.3 | 29.7 | 10.6% | 69.8% | 5.3 | 27.1 | 7.8% |
| | PCT & EMS | 75.8% | 4.4 | 29.3 | 0.3% | **86.0%** | **1.9** | **33.0** | **0.0%** | 75.5% | 4.7 | 29.2 | 0.3% |
| | PCT & EV | **76.0%** | **4.2** | **29.4** | **0.0%** | 85.3% | 2.1 | 32.8 | 0.8% | **75.7%** | **4.6** | **29.2** | **0.0%** |
| | PCT & EVF | 75.7% | 4.8 | 29.2 | 0.4% | 80.5% | 2.9 | 31.0 | 6.4% | 73.5% | 4.6 | 28.4 | 2.9% |
| | PCT & EV/GS | 75.8% | 4.7 | 29.2 | 0.3% | 84.8% | 2.1 | 32.6 | 1.4% | 75.5% | 4.8 | 29.1 | 0.3% |
| | *Random* & EV | 45.7% | 13.5 | 18.4 | 39.9% | 51.0% | 8.3 | 20.4 | 40.7% | 45.1% | 12.5 | 18.1 | 40.4% |

[1] https://github.com/alexfrom0815/Online-3D-BPP-PCT

Although PCT grows under the guidance of heuristics, the combinations of PCT with EMS and EV still learn effective and robust policies outperforming all baselines by a large margin regarding all settings. Note that the closer the space utilization is to 1, the more difficult online 3D-BPP is. It is interesting to see that policies guided by EMS and EV even exceed the performance of the full coordinate space (FC) which is expected to be optimal. This demonstrates that a good leaf node expansion scheme reduces the complexity of the problem and helps DRL agents learn better performance. To prove that the interception of $\mathbf{L}_t$ will not harm the final performance, we train agents with full leaf nodes derived from the EV scheme (EVF) and the test performance is slightly worse than the intercepted cases. We conjecture that the interception keeps the final performance may be caused by two reasons. First, sub-optimal solutions for online 3D-BPP may exist even in the intercepted leaf node set $\mathbf{L}_{sub}$. In addition, the randomly chosen leaf nodes force the agent to make new explorations in case the policy $\pi$ falls into the local optimum. We remove GS from EV cases to prove its effectiveness. The performance of Zhao et al. (2022) deteriorates quickly in $setting\,2$ and $setting\,3$ due to the multiplying orientation space and insufficient state representation separately. Running costs, scalability performance, behavior understanding, visualized results, and the implementation details of our real-world packing system can all be found in Appendix C.

We also repeat the same experiment as Zhang et al. (2021) which pack items sampled from a pre-defined item set $|\mathcal{I}| = 64$ in $setting\,2$. While Zhang et al. (2021) packs on average 15.6 items and achieves $67.0\%$ space utilization, our method packs 19.7 items with a space utilization of $83.0\%$.

## 4.2 BENEFITS OF TREE PRESENTATION

Here we verify that the PCT representation does help online 3D-BPP tasks. For this, we embed each space configuration node independently like PointNet (Qi et al., 2017) to prove the node spatial relations help the final performance. We also deconstruct the tree structure into node sequences and embed them with Ptr-Net (Vinyals et al., 2015), which selects a member from serialized inputs, for indicating that the graph embedding fashion fits our tasks well. We have verified that an appropriate choice of $\mathbf{L}_t$ makes DRL agents easy to train, then we remove the internal nodes $\mathbf{B}_t$ from $\mathcal{T}_t$, along with its spatial relations with other nodes, to prove $\mathbf{B}_t$ is also a necessary part. We choose EV as the leaf node expansion scheme here. The comparison results are summarized in Table 2.

Table 2: A graph embedding for complete PCT helps the final performance.

| Presentation | $Setting\,1$ | | | | $Setting\,2$ | | | | $Setting\,3$ | | | |
| --- | --- | --- | --- | --- | --- | --- | --- | --- | --- | --- | --- | --- |
| | Uti. | Var. | Num. | Gap | Uti. | Var. | Num. | Gap | Uti. | Var. | Num. | Gap |
| PointNet | 69.2% | 6.7 | 26.9 | 8.9% | 78.9% | 3.2 | 30.5 | 7.5% | 71.5% | 5.3 | 27.7 | 5.5% |
| Ptr-Net | 64.1% | 10.0 | 25.1 | 15.7% | 77.5% | 4.1 | 30.1 | 9.1% | 63.5% | 7.9 | 24.8 | 16.1% |
| PCT ($\mathcal{T}/\mathbf{B}$) | 70.9% | 5.9 | 27.5 | 6.7% | 84.1% | 2.6 | 32.3 | 1.4% | 70.6% | 5.3 | 27.4 | 6.7% |
| PCT ($\mathcal{T}$) | **76.0%** | **4.2** | **29.4** | **0.0%** | **85.3%** | **2.1** | **32.8** | **0.0%** | **75.7%** | **4.6** | **29.2** | **0.0%** |

If we ignore the spatial relations between the PCT nodes or only treat the state input as a flattened sequence, the performance of the learned policies will be severely degraded. The presence of $\mathbf{B}$ functions more on $setting\,1$ and $setting\,3$ since $setting\,2$ allows items to be packed in any empty spaces without considering constraints with internal nodes. This also confirms that a complete PCT representation is essential for online 3D-BPP of practical needs.

## 4.3 PERFORMANCE ON CONTINUOUS DATASET

The most concerning issue about online 3D-BPP is their solution space limit. Given that most learning-based methods can only work in a limited discrete coordinate space, we directly test our method in a continuous bin with sizes $S^d = 1$ to prove our superiority. Due to the lack of public datasets for online 3D-BPP issues, we generate item sizes through a uniform distribution $s^d \sim U(a, S^d/2)$, $a$ is set to 0.1 in case endless items are generated. Specifically, for 3D-BPP instances where stability is considered, the diversity of item size $s^z$ needs to be controlled. If all subsets of $\mathbf{B}_t$ meet:

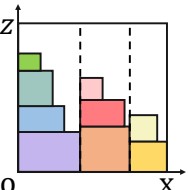

$$\sum_{i \in \mathbf{B}_{sub1}} s_i^z \neq \sum_{i \in \mathbf{B}_{sub2}} s_i^z \quad \mathbf{B}_{sub1} \neq \mathbf{B}_{sub2}, \mathbf{B}_{sub1}, \mathbf{B}_{sub2} \in \mathbf{B}_t \tag{7}$$

This means any packed items cannot form a new plane for providing support in the direction of gravity and the available packing areas shrink. Excessive diversity of $s^z$ will degenerate 3D-BPP into 1D-BPP as shown in the toy demo. To prevent this degradation from leading to the underutilization of the bins, we sample $s^z$ from a finite set $\{0.1, 0.2, \ldots, 0.5\}$ on $setting\,1$ and $setting\,3$.

Table 3: Online 3D-BPP with continuous solution space.

| | Method | Setting 1 | | | | Setting 2 | | | | Setting 3 | | | |
|---|---|---|---|---|---|---|---|---|---|---|---|---|---|
| | | Uti. | Var. | Num. | Gap | Uti. | Var. | Num. | Gap | Uti. | Var. | Num. | Gap |
| Heu. | BR | 40.9% | 7.4 | 16.1 | 37.5% | 45.3% | 5.2 | 17.8 | 31.7% | 40.9% | 7.3 | 16.1 | 38.6% |
| | Ha et al. | 43.9% | 14.2 | 17.2 | 32.9% | 46.1% | 6.8 | 18.1 | 30.5% | 43.9% | 14.2 | 17.2 | 34.1% |
| | LSAH | 48.3% | 12.1 | 18.7 | 26.1% | 58.7% | 4.6 | 22.8 | 11.5% | 48.4% | 12.2 | 18.8 | 27.3% |
| DRL | GD | 5.6% | – | 2.2 | 91.4% | 7.5% | – | 2.9 | 88.7% | 5.2% | – | 2.1 | 92.2% |
| | PCT & EMS | 65.3% | 4.4 | 24.9 | 0.2% | **66.3%** | **2.3** | **27.0** | **0.0%** | **66.6%** | **3.3** | **25.3** | **0.0%** |
| | PCT & EV | **65.4%** | **3.3** | **25.0** | **0.0%** | 65.0% | 2.6 | 26.4 | 2.0% | 65.8% | 3.6 | 25.1 | 2.7% |

We find that some heuristic methods (Ha et al., 2017) also have the potential to work in the continuous domain. We improve these methods as our baselines. Another intuitive approach for solving online 3D-BPP with continuous solution space is driving a DRL agent to sample actions from a gaussian distribution (GD) and output continuous coordinates directly. The test results are summarized in Table 3. Although the continuous-size item set is infinite ($|\mathcal{I}| = \infty$) which increases the difficulty of the problem and reduces the performance of all methods, our method still performs better than all competitors. The DRL agent which outputs continuous actions directly cannot even converge and their variance is not considered. Our work is, to our knowledge, the first that deploys the learning-based method on solving online 3D-BPP with continuous solution space successfully.

## 4.4 GENERALIZATION ON DIFFERENT DISTRIBUTIONS

The generalization ability of learning-based methods has always been a concern. Here we demonstrate that our method has a good generalization performance on item size distributions different from the training one. We conduct this experiment with continuous solution space. We sample $s^d$ from normal distributions $N(\mu, \sigma^2)$ for generating test sequences where $\mu$ and $\sigma$ are the expectation and the standard deviation. Three normal distributions are adopted here, namely $N(0.3, 0.1^2)$, $N(0.1, 0.2^2)$, and $N(0.5, 0.2^2)$, as shown in Figure 3. The larger $\mu$ of the normal distribution, the larger the average size of sampled items. We still control $s^d$ within the range of $[0.1, 0.5]$. If the sampled item sizes are not within this range, we will resample until it meets the condition.

Table 4: Generalization performance on different kinds of item sampling distributions.

| Test Distribution | Method | Setting 1 | | | Setting 2 | | | Setting 3 | | |
|---|---|---|---|---|---|---|---|---|---|---|
| | | Uti. | Var. | Num. | Uti. | Var. | Num. | Uti. | Var. | Num. |
| $s^d \sim U(0.1, 0.5)$ | LSAH | 48.3% | 12.1 | 18.7 | 58.7% | 4.6 | 22.8 | 48.4% | 12.2 | 18.8 |
| | PCT & EMS | 65.3% | 4.4 | 24.9 | **66.3%** | **2.3** | **27.0** | **66.6%** | **3.3** | **25.3** |
| | PCT & EV | **65.4%** | **3.3** | **25.0** | 65.0% | 2.6 | 26.4 | 65.8% | 3.6 | 25.1 |
| $s^d \sim N(0.3, 0.1^2)$ | LSAH | 49.2% | 11.1 | 18.9 | 60.0% | 4.1 | 22.9 | 49.2% | 11.0 | 18.9 |
| | PCT & EMS | **66.1%** | 3.6 | **25.1** | **64.3%** | 3.5 | **25.6** | **66.4%** | 3.0 | **25.2** |
| | PCT & EV | 65.1% | **2.8** | 24.7 | 63.7% | **2.6** | 25.3 | 66.2% | **2.9** | 25.1 |
| $s^d \sim N(0.1, 0.2^2)$ | LSAH | 52.4% | 8.9 | 30.3 | 62.9% | **2.4** | 44.3 | 52.3% | 8.9 | 30.2 |
| | PCT & EMS | **68.5%** | 2.5 | **39.0** | **66.4%** | 3.0 | **49.7** | **69.2%** | 2.5 | **39.4** |
| | PCT & EV | 66.5% | 2.7 | 38.0 | 64.9% | 2.7 | 48.3 | 67.4% | **2.4** | 38.5 |
| $s^d \sim N(0.5, 0.2^2)$ | LSAH | 47.3% | 12.6 | 13.0 | 56.0% | 5.5 | 12.9 | 47.3% | 12.6 | 13.0 |
| | PCT & EMS | 63.5% | 5.0 | 17.3 | **64.5%** | **2.8** | **15.4** | 65.2% | 3.8 | **17.7** |
| | PCT & EV | **65.1%** | **3.3** | **17.7** | 64.5% | 2.8 | 15.3 | 65.1% | **3.7** | 17.7 |

We directly transfer our policies trained on $U(0.1, 0.5)$ to these new datasets without any fine-tuning. We use the best-performing heuristic method LSAH (Hu et al., 2017) in Section 4.3 as a baseline. The test results are summarized in Table 4. Our method performs well on the distributions different from the training one and always surpasses the LSAH method. See Appendix C for more results about the generalization ability of our method on disturbed distributions and unseen items.

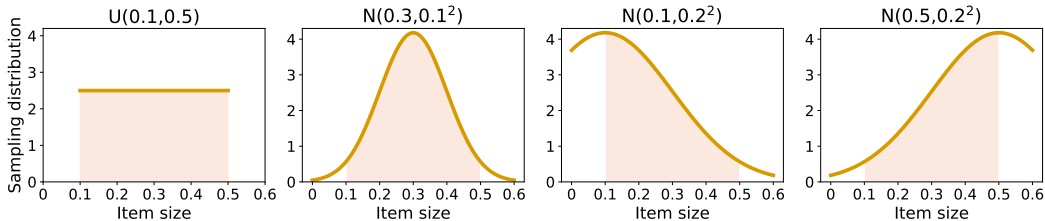

Figure 3: The probability distribution for sampling item sizes. The area of the colored zone is normalized to 1.

### 4.5 MORE COMPLEX PRACTICAL CONSTRAINTS

To further prove that our method fits 3D-BPP with complex constraints well, we give two demonstrations about extending our method to online 3D-BPP with additional practical constraints: isle friendliness and load balancing (Gzara et al., 2020):

**3D-BPP with Isle Friendliness** Isle friendliness refers to that items belonging to the same category should be packed as closely as possible. The item weight is set as $w_t = max(0, v_t - c \cdot dist(n_t, \mathbf{B}_t))$, $c$ is a constant. The additional object function $dist(n_t, \mathbf{B}_t)$ means the average distance between $n_t$ and the items of the same category in $\mathbf{B}_t$. The additional category information is appended into the descriptors of $\mathbf{B}_t$ and $n_t$. Four item categories are tested here.

**3D-BPP with Load Balancing** Load balancing dictates that the packed items should have an even mass distribution within the bin. The item weight is set as $w_t = max(0, v_t - c \cdot var(n_t, \mathbf{B}_t))$. Object $var(n_t, \mathbf{B}_t)$ is the variance of the mass distribution of the packed items on the bottom of the bin.

Table 5: Online 3D-BPP with practical constraints. Obj. means the task-specific objective score, the smaller the better. $Setting\,2$ involves no mass property and the load balancing is not considered.

| | Method | $Setting\,1$ | | | $Setting\,2$ | | | $Setting\,3$ | | |
|---|---|---|---|---|---|---|---|---|---|---|
| | | Uti. | Num. | Obj. | Uti. | Num. | Obj. | Uti. | Num. | Obj. |
| Isle Friendliness | Zhao et al. | 58.3% | 22.5 | 4.58 | 64.2% | 24.8 | 4.67 | 59.0% | 22.8 | 4.60 |
| | PCT & EV | 72.1% | 29.0 | 4.44 | 85.2% | 32.8 | 2.69 | 74.6% | 28.8 | 4.45 |
| Load Balancing | Zhao et al. | 60.3% | 23.3 | 88.0 | | — | | 61.1% | 23.7 | 30.1 |
| | PCT & EV | 73.7% | 28.6 | 40.5 | | — | | 74.0% | 28.7 | 20.9 |

We choose the learning-based method Zhao et al. (2022) as our baseline since heuristic methods only take space utilization as their objects. The results are summarized in Table 5. Compared with the baseline algorithm, our method better achieves the additional object while still taking into account the space utilization as the primary.

## 5 DISCUSSION

We formulate the online 3D-BPP as a novel hierarchical representation -– packing configuration tree. PCT is a full-fledged description of the state and action space of bin packing which makes the DRL agent easy to train and well-performing. We extract state feature from PCT using graph attention networks which encodes the spatial relations of all space configuration nodes. The graph representation of PCT helps the agent with handling online 3D-BPP with complex practical constraints, while the finite leaf nodes prevent the action space from growing explosively. Our method surpasses all other online 3D-BPP algorithms and is the first learning-based method that solves online 3D-BPP with continuous solution space successfully. Our method performs well even on item sampling distributions different from the training one. We also give demonstrations to prove that our method is versatile in terms of incorporating various practical constraints. For future research, we are extremely interested in applying our method to the more difficult irregular shape packing (Wang & Hauser, 2019a), where the sampling costs for training DRL agents are more expensive and the solution space will be far more complex. A good strategy for balancing exploration and exploitation for learning agents is eagerly needed in this situation. Finding other alternatives for better representing 3D-BPP with different packing constraints and designing better leaf node expansion schemes for PCT are also interesting directions.

## ACKNOWLEDGEMENTS

We thank Zhan Shi, Wei Yu, and Lintao Zheng for helpful discussions. We would also like to thank anonymous reviewers for their insightful comments and valuable suggestions. This work is supported by the National Natural Science Foundation of China (62132021, 61876077).

## ETHICS STATEMENT

We believe the broader impact of this research is significant. BPP is considered one of the most needed academic problems due to its wide applicability in our daily lives (Skiena, 1997). Providing a good solution to online 3D-BPP yields direct and profound impacts beyond academia. We note that online packing setups have been already widely deployed in logistics hubs and manufacturing plants. A good online 3D-BPP policy benefits all the downstream operations like package wrapping, transportation, warehousing, and distribution.

Our method can also be treated as a separate decision module and utilized to solve other variants of 3D-BPP issues, such as multi-bin online 3D-BPP, online 3D-BPP with lookahead (Zhao et al., 2021), online placement decision for offline 3D-BPP (Hu et al., 2017), and the logistics robot implementation for online packing (Zhao et al., 2022), which have been well studied in previous works. We give demonstrations about applying our method to solving online 3D-BPP with various complex practical constraints, the similar idea can also be adopted to meet other customized desires.

## REPRODUCIBILITY

Our real-world video demo is submitted with the supplemental material. The source code of our method and related baselines is published at `https://github.com/alexfrom0815/Online-3D-BPP-PCT`.

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

In this appendix, we provide more details and statistical results of our PCT method. Our real-world packing demo is also submitted with the supplemental material.

- Section A gives more descriptions about training methods, which include the implementation of the deep reinforcement learning method, specific GAT network designs for extracting problem features from PCT, and recommendations for finding suitable PCT length.

- Section B elaborates the concept of the leaf node expansion schemes adopted for finding candidate placements in our method. Learning curves and the computational complexity analysis of these schemes are also provided in this section.

- Section C reports more statistical results of our method, including further discussions on generalization ability, running costs, and scalability. The understanding of model behaviors, the visualized results, and details of our real-world packing system are also provided.

## A    IMPLEMENTATION DETAILS

**Deep Reinforcement Learning**    We formulate online 3D-BPP as Markov Decision Process and solve it with the deep reinforcement learning method. A DRL agent seeks for a policy $\pi$ to maximize the accumulated discounted reward:

$$J(\pi) = E_{\tau \sim \pi}[\sum_{t=0}^{\infty} \gamma^t R(s_t, a_t)] \tag{8}$$

Where $\gamma \in [0, 1]$ is the discount factor, and $\tau = (s_0, a_0, s_1, \ldots)$ is a trajectory sampled based on the policy $\pi$. We extract the feature of state $s_t = (\mathcal{T}_t, n_t)$ using graph attention networks (Velickovic et al., 2018) for encoding the spatial relations of all space configuration nodes. The context feature is fed to two key components of our pipeline: an actor network and a critic network. The actor network, designed based on pointer mechanism, weighs the leaf nodes of PCT, which is written as $\pi(a_t|s_t)$. The action $a_t$ is an index of selected leaf node $l \in \mathbf{L}_t$, denoted as $a_t = index(l)$. The critic network maps the context feature into a state value prediction $V(s_t)$ which helps the training of actor network. The whole network is trained via a composite loss $L = \alpha \cdot L_{actor} + \beta \cdot L_{critic}$ ($\alpha = \beta = 1$ in our implementation), which consists of actor loss $L_{actor}$ and critic loss $L_{critic}$. These two loss functions are defined as:

$$\begin{cases} L_{actor} & = (r_t + \gamma V(s_{t+1}) - V(s_t)) \log \pi(a_t|s_t) \\ L_{critic} & = (r_t + \gamma V(s_{t+1}) - V(s_t))^2 \end{cases} \tag{9}$$

Where $r_t = c_r \cdot w_t$ is our reward signal and we set $\gamma$ as 1 since the packing episode is finite. We adopt a step-wise reward $r_t = c_r \cdot w_t$ once $n_t$ is inserted into PCT as an internal node successfully. Otherwise, $r_t = 0$ and the packing episode ends. The choice of item weight $w_t$ depends on the packing preferences. In the general sense, we set $w_t$ as the volume occupancy $v_t = s_n^x \cdot s_n^y \cdot s_n^z$ of $n_t$, and the constant $c_r$ is $10/(S^x \cdot S^y \cdot S^z)$. For online 3D-BPP with additional packing constraints, this weight can be set as $w_t = max(0, v_t - c \cdot O(s_t, a_t))$. While term $v_t$ ensures that space utilization is still the primary concern, the objective function $O(s_t, a_t)$ guides the agent to satisfy additional constraints like isle friendliness and load balancing. We adopt ACKTR (Wu et al., 2017) method for training our DRL agent, which iteratively updates an actor and a critic using Kronecker-factored approximate curvature (K-FAC) (Martens & Grosse, 2015) with trust region. Zhao et al. (2021) have demonstrated that this method has a surprising superiority on online 3D packing problems over other model-free DRL algorithms like SAC (Haarnoja et al., 2018).

**Feature extraction**    Specifically, ACKTR runs multiple parallel processes (64 here) to interact with their respective environments and gather samples. The different processes may have different packing time step $t$ and deal with different packing sequences, the space configuration nodes number N also changes. To combine these data with irregular shapes into one batch, we fullfill $\mathbf{B}_t$ and $\mathbf{L}_t$ to fixed lengths, 80 and $25 \cdot |\mathbf{O}|$ respectively, with dummy nodes. The descriptors for dummy nodes are all-zero vectors and have the same size as the internal nodes or the leaf nodes. The relation weight logits $u_{ij}$ of dummy node $j$ to arbitrary node $i$ is replaced with $-inf$ to eliminate these dummy nodes during the feature calculation of GAT. The global context feature $\bar{h}$ is aggregated only on the eligible nodes $\mathbf{h}$: $\bar{h} = \frac{1}{N} \sum_{i=1}^{N} h_i$. All space configuration nodes are embedded by GAT as a fully connected graph as Figure 2 (b), without any inner mask operation.

We only provide the packable leaf nodes which satisfy placement constraints for DRL agents. For $setting\,2$, we check in advance if a candidate placement satisfies Constraint 1 and 2. For $setting\,1$ and $setting\,3$ where the mass of item $n_t$ is $v_t$ and $\rho \cdot v_t$ respectively, we will additionally check if one placement meets the constraints of packing stability. Benefits from the fast stability estimation method proposed by Zhao et al. (2022), this pre-checking process can be completed in a very short time, and our DRL agent samples data at a frequency of more than 400 FPS.

The node-wise MLPs $\phi_{\theta_B}, \phi_{\theta_L}$, and $\phi_{\theta_n}$ used to embed raw space configuration nodes are two-layer linear networks with LeakyReLU activation function. $\phi_{FF}$ is a two-layer linear structure activated by ReLU. The feature dimensions $d_h, d_k$, and $d_v$ are 64. The hyperparameter $c_{clip}$ used to control the range of clipped compatibility logits is set to 10 in our GAT implementation.

**Choice of PCT Length**  Since PCT allows discarding some valid leaf nodes and this will not harm our performance, we randomly intercept a subset $\mathbf{L}_{sub_t}$ from $\mathbf{L}_t$ if $|\mathbf{L}_t|$ exceeds a certain length. Determining the suitable PCT length for different bin configurations is important, we give our recommendations for finding this hyperparameter. For training, we find that the performance of learned policies is more sensitive to the number of allowed orientations $|\mathbf{O}|$. Thus we set the PCT length as $c \cdot |\mathbf{O}|$ where $c$ can be determined by a grid search nearby $c = 25$ for different bin configurations. For our experiments, $c = 25$ works quite well. During the test, the PCT length can be different from the training one, we suggest searching this interception length with a validation dataset via a grid search which ranges from 50 to 300 with step length 10.

## B  LEAF NODE EXPANSION SCHEMES

We introduce the leaf node expansion schemes adopted in our PCT implementation here. These schemes are used to incrementally calculate new candidate placements introduced by the just placed item $n_t$. A good expansion scheme should reduce the number of solutions to be explored while not missing too many feasible packings. Meanwhile, polynomially computability is also expected. As shown in Figure 4, the policies guided by suitable leaf node expansion schemes outperform the policy trained on a full coordinate (FC) space in the whole training process. We extend three existing heuristic placement rules which have proven to be both accurate and efficient to our PCT expansion, i.e. *Corner Point*, *Extreme Point*, and *Empty Maximal Space*. Since all these schemes are related to boundary points of packed items, we combine the start/end points of $n_t$ with these boundary points as a superset, namely *Event Point*.

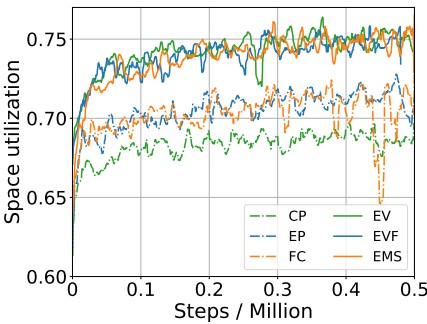

Figure 4: Learning curves on $setting\,1$. A good expansion scheme for PCT reduces the complexity and helps DRL methods for more efficient learning and better performance. EVF means the full EV leaf node set without an interception.

**Corner Point**  Martello et al. (2000) first introduce the concept of Corner Point (CP) for their branch-and-bound methods. Given 2D packed items in the $xoy$ plane, the corner points can be found where the envelope of the items in the bin changes from vertical to horizontal, as shown in Figure 5 (a). The past corner points which no longer meet this condition will be deleted. Extend this 2D situation to 3D cases, the new candidate 3D positions introduce by the just placed item $n_t$ are a subset of $\{(p_n^x + s_n^x, p_n^y, p_n^z), (p_n^x, p_n^y + s_n^y, p_n^z), (p_n^x, p_n^y, p_n^z + s_n^z)\}$ if the envelope of the corresponding 2D plane, i.e. $xoy, yoz$, and $xoz$, is changed by $n_t$. The time complexity of finding 3D corner points incrementally is $O(c)$ with an easy-to-maintained bin height map data structure to detect the change of envelope on each plane, $c$ is a constant here.

**Extreme Point**  Crainic et al. (2008) extend the concept of Corner Point to Extreme Point (EP) and claim their method reaches the best offline performance of that era. Its insight is to provide the means to exploit the free space defined inside a packing by the shapes of the items that already exist. When the current item $n_t$ is added, new EPs are incrementally generated by projecting the coordinates $\{(p_n^x + s_n^x, p_n^y, p_n^z), (p_n^x, p_n^y + s_n^y, p_n^z), (p_n^x, p_n^y, p_n^z + s_n^y)\}$ on the orthogonal axes, e.g., project $(p_n^x + s_n^x, p_n^y, p_n^z)$ in the directions of the $y$ and $z$ axes to find intersections with all items lying between item $n_t$ and the boundary of the bin. The nearest intersection in the respective direction is an

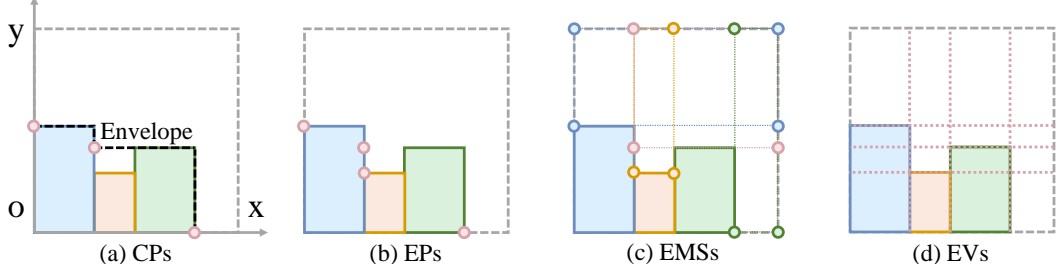

Figure 5: Full candidate positions generated by different PCT expansion schemes (all in $xoy$ plane). The gray dashed lines are the boundaries of the bin. Circles in (a) and (b) represent corner points and extreme points respectively. (c): The candidate positions (circles) introduced by different EMSs are rendered with different colors. All intersections of two dashed lines in (d) constitute event points.

extreme point. Since we stipulate a vertical top-down loading direction, the 3D extreme points in the strict sense may exist a large item blocking the loading direction. So we find the 2D extreme points (see Figure 5 (b)) in the $xoy$ plane and repeat this operation on each distinct $p^z$ value (i.e. start/end $z$ coordinate of a packed item) which satisfies $p_n^z \leq p^z \leq p_n^z + s_n^z$. The time complexity of this method is $O(m \cdot |\mathbf{B}_{2D}|)$, where $\mathbf{B}_{2D}$ is the packed items that exist in the corresponding $z$ plane and $m$ is the number of related $z$ scans.

**Empty Maximal Space** Empty Maximal Spaces (EMSs) (Ha et al., 2017) are the largest empty orthogonal spaces whose sizes cannot extend more along the coordinate axes from its front-left-bottom (FLB) corner. This is a simple and effective placement rule. An EMS $e$ is presented by its FLB corner $(p_e^x, p_e^y, p_e^z)$ and sizes $(s_e^x, s_e^y, s_e^z)$. When the current item $n_t$ is placed into $e$ on its FLB corner, this EMS is split into three smaller EMSs with positions $(p_e^x + s_n^x, p_e^y, p_e^z)$, $(p_e^x, p_e^y + s_n^y, p_e^z)$, $(p_e^x, p_e^y, p_e^z + s_n^z)$ and sizes $(s_e^x - s_n^x, s_e^y, s_e^z)$, $(s_e^x, s_e^y - s_n^y, s_e^z)$, $(s_e^x, s_e^y, s_e^z - s_n^z)$, respectively. If the item $n_t$ only partially intersects with $e$, we can apply a similar volume subtraction to the intersecting part for splitting $e$. For each ems, we define the left-up $(p_e^x, p_e^y + s_e^y, p_e^z)$, right-up $(p_e^x + s_e^x, p_e^y + s_e^y, p_e^z)$, left-bottom $(p_e^x, p_e^y, p_e^z)$, and right-bottom $(p_e^x + s_e^x, p_e^y, p_e^z)$ corners of its vertical bottom as candidate positions, as shown in Figure 5 (c). These positions also need to be converted to the FLB corner coordinate for placing item $n_t$. The left-up, right-up, left-bottom and right-bottom corners of $e$ should be converted to $(p_e^x, p_e^y + s_e^y - s_n^y, p_e^z)$, $(p_e^x + s_e^x - s_n^x, p_e^y + s_e^y - s_n^y, p_e^z)$, $(p_e^x, p_e^y, p_e^z)$, and $(p_e^x + s_e^x - s_n^x, p_e^y, p_e^z)$ respectively. Since all EMSs $e \in \mathbf{E}$ in the bin needs to detect intersection with $n_t$, the time complexity of finding 3D EMSs incrementally is $O(|\mathbf{E}|)$. A 3D schematic diagram of PCT expansion guided by EMSs is provided in Figure 6.

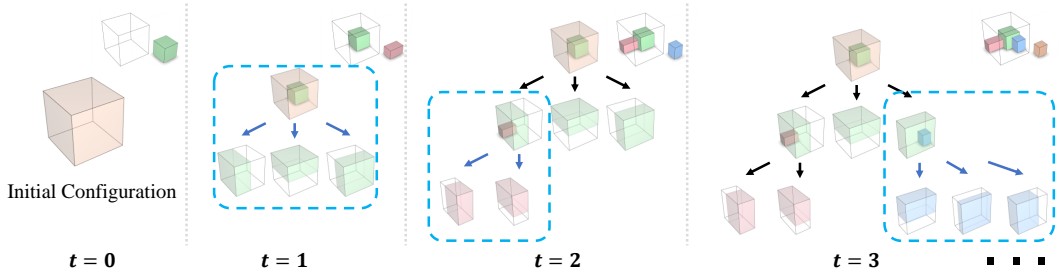

Figure 6: A 3D PCT expansion schematic diagram. This PCT grows under the guidance of the EMS expansion scheme. For simplicity, we only choose the bottom-right-up corners of each EMS as candidate positions and we set $|\mathbf{O}| = 1$ here.

**Event Point** It's not difficult to find that all schemes mentioned above are related to boundary points of a packed item along $d \in \{x, y\}$ axes (we assume the initial empty bin is also a special packed item here). When the current item $n_t$ is packed, we update the existing PCT leaf nodes by scanning all distinct $p^z$ values which satisfy $p_n^z \leq p^z \leq p_n^z + s_n^z$ and combine the start/end points of $n_t$ with the boundary points that exist in this $z$ plane to get the superset (see Figure 5 (d)), which is called *Event Points*. The time complexity for detecting event points is $O(m \cdot |B_{2D}|^2)$.

## C MORE RESULTS

In this section, we report more results of our method. Section C.1 further discusses the generalization ability of our method on disturbed item sampling distributions and unseen items. Section C.2 reports the running cost of each method. Section C.3 scales our method to a larger problem and reports the performance. Section C.4 visualizes packing sequences to analyze model behavior and more visualized results are provided in Section C.5. Section C.6 introduces our real-world packing system.

### C.1 FURTHER DISCUSSIONS ON GENERALIZATION

We have verified in Section 4.4 that our method has good generalization performance on item sampling distributions different from the training one. Now we further analyze the generalization ability of our method. Firstly, we demonstrate that our algorithm has a good generalization performance on disturbed item sampling distributions. We conduct this experiment in the discrete setting where the item set is finite ($|\mathcal{I}| = 125$). For each item $i \in \mathcal{I}$, we add a random non-zero disturbance $\delta_i$ on its original sample probability $p_i$, e.g., $p_i = p_i \cdot (1 - \delta_i)$. We normalize the disturbed $p_i$ as the final item sampling probability. Note that $\delta_i$ is fixed during sampling one complete sequence. The state transition of DRL $\mathcal{P}(s_{t+1}|s_t)$ which is partly determined by the probabilities of sampling items will be different from the training transition where the uniform distribution sample items with equal probabilities. Generalizing to a new transition is a classic challenge for reinforcement learning (Taylor & Stone, 2009). We test different ranges of $\delta_i$ and the results are summarized in Table 6.

Table 6: Transfer the best-performing PCT policies directly to the disturbed item sampling distributions. Dif. means how much the generalization performance drops from the undisturbed case ($\delta_i = 0$).

| Disturbance | Setting 1 | | | | Setting 2 | | | | Setting 3 | | | |
| --- | --- | --- | --- | --- | --- | --- | --- | --- | --- | --- | --- | --- |
| | Uti. | Var. | Num. | Dif. | Uti. | Var. | Num. | Dif. | Uti. | Var. | Num. | Dif. |
| $\delta_i = 0$ | 76.0% | 4.2 | 29.4 | 0.0% | 86.0% | 1.9 | 33.0 | 0.0% | 75.7% | 4.6 | 29.2 | 0.0% |
| $\delta_i \in [-20\%, 20\%]$ | 75.6% | 4.6 | 29.1 | 0.5% | 85.7% | 2.1 | 32.8 | 0.3% | 75.3% | 4.5 | 29.0 | 0.5% |
| $\delta_i \in [-40\%, 40\%]$ | 75.5% | 4.5 | 29.0 | 0.7% | 85.6% | 2.1 | 32.8 | 0.5% | 75.6% | 4.8 | 29.3 | 0.1% |
| $\delta_i \in [-60\%, 60\%]$ | 75.5% | 4.3 | 28.9 | 0.7% | 85.8% | 2.1 | 32.8 | 0.2% | 75.5% | 4.8 | 28.9 | 0.3% |
| $\delta_i \in [-80\%, 80\%]$ | 75.7% | 4.5 | 29.2 | 0.4% | 85.6% | 2.2 | 32.9 | 0.5% | 75.4% | 4.9 | 29.3 | 0.4% |
| $\delta_i \in [-100\%, 100\%]$ | 75.8% | 4.4 | 29.0 | 0.3% | 85.5% | 2.2 | 32.6 | 0.6% | 75.3% | 4.7 | 29.3 | 0.5% |

Benefits from the efficient guidance of heuristic leaf node expansion schemes, our method maintains its performance under various amplitude disturbances. Our method even behaves well with a strong disturbance $\delta_i \in [-100\%, 100\%]$ applied, which means some items may never be sampled by some distributions when $\delta_i = 1$ and $p_i \cdot (1 - \delta_i) = 0$ in a specific sequence.

Beyond experiments on generalization to disturbed distributions, we also test our method with unseen items. We conduct this experiment in the discrete setting. We randomly delete 25 items from $\mathcal{I}$ and train PCT policies with $|\mathcal{I}_{sub}| = 100$. Then we test the trained policies on full $\mathcal{I}$. See Table 7 for results. Our method still performs well on datasets where unseen items exist regarding all settings.

Table 7: Generalization performance on unseen items. All policies are trained with the EV scheme.

| Train | Test | Setting 1 | | | Setting 2 | | | Setting 3 | | |
| --- | --- | --- | --- | --- | --- | --- | --- | --- | --- | --- |
| | | Uti. | Var. | Num. | Uti. | Var. | Num. | Uti. | Var. | Num. |
| $|\mathcal{I}| = 125$ | $|\mathcal{I}| = 125$ | 76.0% | 4.2 | 29.4 | 85.3% | 2.1 | 32.8 | 75.7% | 4.6 | 29.2 |
| $|\mathcal{I}_{sub}| = 100$ | $|\mathcal{I}_{sub}| = 100$ | 74.4% | 5.1 | 29.4 | 86.3% | 1.7 | 33.8 | 74.2% | 4.7 | 29.3 |
| $|\mathcal{I}_{sub}| = 100$ | $|\mathcal{I}| = 125$ | 74.6% | 5.4 | 28.9 | 85.6% | 2.6 | 33.0 | 74.4% | 5.2 | 28.8 |

### C.2 RUNNING COSTS

For 3D-BPP of online needs, the running cost for placing each item is especially important. We count the running costs of the experiments in Section 4.1 and Section 4.3 and summarize them in Table 8. Each running cost at time step $t$ is counted from putting down the previous item $n_{t-1}$ until the current item $n_t$ is placed, which includes the time to make placement decisions, the time to check placement feasibility, and the time to interact with the packing environment. The running cost of our method is comparable to most baselines. Our method can meet real-time packing requirements in both discrete solution space and continuous solution space.

Table 8: Running costs (*seconds*) tested on online 3D-BPP with discrete solution space (Section 4.1) and continuous solution space (Section 4.3). The running costs of the latter are usually more expensive since checking Constraint 1 and 2 in the continuous domain is more time-consuming.

| | Method | Setting 1 | | Setting 2 | | Setting 3 | |
|---|---|---|---|---|---|---|---|
| | | Discrete | Continuous | Discrete | Continuous | Discrete | Continuous |
| Heuristic | *Random* | $4.59 \times 10^{-2}$ | – | $2.03 \times 10^{-2}$ | – | $4.62 \times 10^{-2}$ | – |
| | Wang & Hauser | $4.76 \times 10^{-2}$ | – | $3.01 \times 10^{-2}$ | – | $4.55 \times 10^{-2}$ | – |
| | DBL | $5.58 \times 10^{-2}$ | – | $1.87 \times 10^{-2}$ | – | $5.44 \times 10^{-2}$ | – |
| | BR | $1.50 \times 10^{-2}$ | $1.69 \times 10^{-2}$ | $1.74 \times 10^{-2}$ | $1.76 \times 10^{-2}$ | $1.42 \times 10^{-2}$ | $1.62 \times 10^{-2}$ |
| | Ha et al. | $5.89 \times 10^{-3}$ | $1.48 \times 10^{-2}$ | $3.39 \times 10^{-3}$ | $7.17 \times 10^{-3}$ | $4.86 \times 10^{-3}$ | $1.38 \times 10^{-2}$ |
| | LSAH | $1.22 \times 10^{-2}$ | $1.44 \times 10^{-2}$ | $4.98 \times 10^{-3}$ | $7.02 \times 10^{-3}$ | $1.14 \times 10^{-2}$ | $1.33 \times 10^{-2}$ |
| | MACS | $2.68 \times 10^{-2}$ | – | $3.00 \times 10^{-2}$ | – | $2.79 \times 10^{-2}$ | – |
| DRL | Zhao et al. | $5.51 \times 10^{-2}$ | – | $1.33 \times 10^{-2}$ | – | $3.31 \times 10^{-2}$ | – |
| | PCT & CP | $8.43 \times 10^{-3}$ | $1.61 \times 10^{-2}$ | $7.36 \times 10^{-3}$ | $1.52 \times 10^{-2}$ | $8.79 \times 10^{-3}$ | $1.73 \times 10^{-2}$ |
| | PCT & EP | $1.22 \times 10^{-2}$ | $3.73 \times 10^{-2}$ | $1.13 \times 10^{-2}$ | $1.57 \times 10^{-2}$ | $1.25 \times 10^{-2}$ | $3.65 \times 10^{-2}$ |
| | PCT & EMS | $1.77 \times 10^{-2}$ | $4.11 \times 10^{-2}$ | $9.49 \times 10^{-3}$ | $2.36 \times 10^{-2}$ | $1.80 \times 10^{-2}$ | $3.08 \times 10^{-2}$ |
| | PCT & EV | $2.66 \times 10^{-2}$ | $4.46 \times 10^{-2}$ | $1.25 \times 10^{-2}$ | $3.21 \times 10^{-2}$ | $2.61 \times 10^{-2}$ | $4.38 \times 10^{-2}$ |

## C.3 SCALABILITY

The number of PCT nodes changes constantly with the generation and removal of leaf nodes during the packing process. To verify whether our method can solve packing problems with a larger scale $|\mathbf{B}|$, we conduct a stress test on *setting* 2 where the most orientations are allowed and the most leaf nodes are generated. We limit the maximum item sizes $s^d$ to $S^d/5$ so that more items can be accommodated. We transfer the best-performing policies on *setting* 2 (trained with EMS) to these new datasets without any fine-tuning. The results are summarized in Table 9.

Table 9: Scalability on larger packing problems. $|\mathbf{L}|$ is the average number of leaf nodes per step. Run. is the running cost. $|\mathbf{L}|$ will not increase exponentially with $|\mathbf{B}|$ since invalid leaf nodes will be removed.

| Item sizes | Discrete | | | | | Continuous | | | | |
|---|---|---|---|---|---|---|---|---|---|---|
| | $\|\mathbf{B}\|$ | $\|\mathbf{L}\|$ | Uti. | Run. | $\|\mathbf{L}\|/\|\mathbf{B}\|$ | $\|\mathbf{B}\|$ | $\|\mathbf{L}\|$ | Uti. | Run. | $\|\mathbf{L}\|/\|\mathbf{B}\|$ |
| $[S^d/10, S^d/2]$ | 33.0 | 51.5 | 86.0% | $9.5 \times 10^{-2}$ | 1.6 | 27.0 | 197.5 | 66.3% | $2.4 \times 10^{-2}$ | 7.3 |
| $[S^d/10, S^d/5]$ | 241.3 | 67.2 | 81.3% | $9.8 \times 10^{-3}$ | 0.3 | 185.4 | 956.5 | 61.9% | $3.7 \times 10^{-2}$ | 5.2 |

PCT size will not grow exponentially with packing scale $|\mathbf{B}|$ since invalid leaf nodes will be removed from leaf nodes $\mathbf{L}$ during the packing process, both discrete and continuous cases. For continuous cases, $|\mathbf{L}|$ is more sensitive to $|\mathbf{B}|$ due to the diversity of item sizes (i.e. $|\mathcal{I}| = \infty$), however, $|\mathbf{L}|$ still doesn't explode with $|\mathbf{B}|$ and it grows in a sub-linear way. Our method can execute packing decisions at a real-time speed with controllable PCT sizes even if the item scale is around two hundred.

## C.4 UNDERSTANDING OF MODEL BEHAVIORS

The qualitative understanding of model behaviors is important, especially for practical concerns. We visualize our packing sequences to give our analysis. The behaviors of learned models differ with the packing constraints. If there is no specific packing preference, our learned policies will start packing nearby a fixed corner (Figure 7 (a)). The learned policies tend to combine items of different heights together to form a plane for supporting future ones (Figure 7 (b)). Meanwhile, it prefers to assign little items to gaps and make room (Figure 7 (c)) for future large ones (Figure 7 (d)). If additional packing preference is considered, the learned policies behave differently. For online 3D-BPP with load balancing, the model will keep the maximum height in the bin as low as possible and pack items layer by layer (Figure 7(e)). For online 3D-BPP with isle friendliness, our model tends to pack the same category of items nearby the same bin corner (Figure 7 (f)).

## C.5 VISUALIZED RESULTS

We visualize the experimental results of Section 4.1 in Figure 8. Each plot is about a randomly generated item sequence packed by our best-performing policies (PCT & EV on *setting* 1 and *setting* 3, PCT & EMS on *setting* 2).

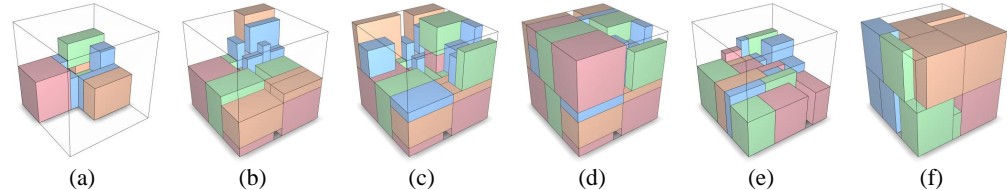

Figure 7: (a)∼(d): Different packing stages of the same sequence. The learned policies assign little items (colored in blue) to gaps and save room for future uncertainty. (e): Online 3D-BPP where load balancing is considered. (f): Online 3D-BPP with isle-friendliness, different color means different item categories.

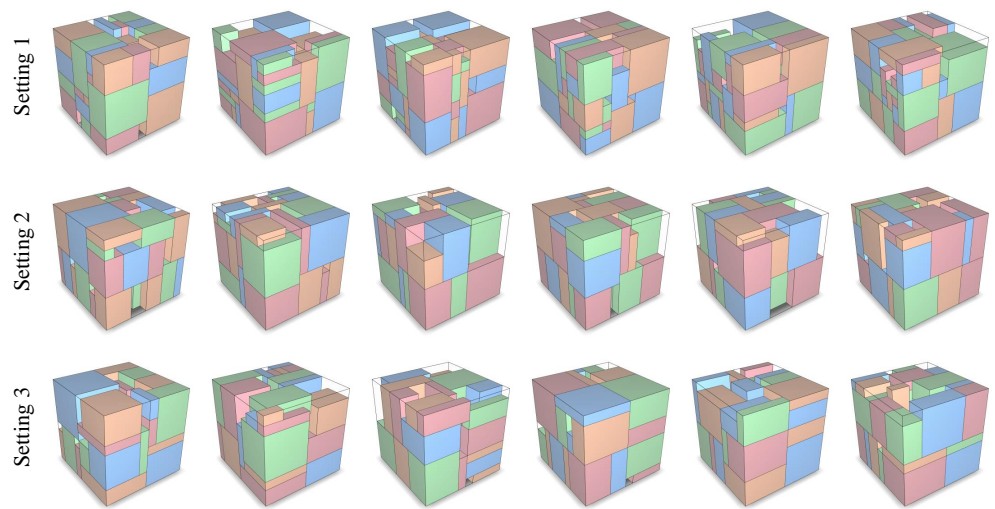

Figure 8: Visualized results of our method.

## C.6 REAL-WORLD PACKING SYSTEM

We follow the online autonomous bin packing system implementation of Zhao et al. (2022) in a logistics warehouse to validate our PCT method. As shown in Figure 9, the conveyor belt transports items to the robot arm (STEP®SR20E) at a fixed speed. The on-conveyor RGB-D camera (PhoXi 3D Scanner XL) recognizes the size and location of the current item. This information is sent to the robot arm for picking the current item and our PCT model for the placement decision. We align the bin coordinate with the robot coordinate and transform the packing decision to a real-world position. Then the robot arm places the current item into the bin which is a table trolley with sizes $S^x = 110cm, S^y = 90cm$, and $S^z = 80cm$. We record sizes and positions of packed items as descriptors of internal nodes **B**.

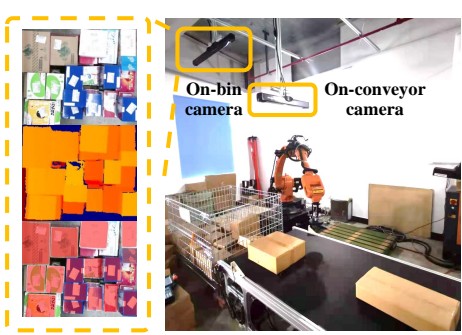

Figure 9: Our packing system. The on-conveyor camera detects packing targets. The on-bin camera monitors possible drifts of packed items.

Different from the original implementation of Zhao et al. (2022) where only one on-conveyor camera is used for detecting the target item to be packed, we add one more on-bin RGB-D camera and adopt Mask R-CNN (He et al., 2017) for instance segmentation of input depth image, for monitoring possible drift of packed items. If a packed item deviates from the position of the PCT decision and this drift is detected, we will correct the descriptor of the corresponding internal node $b \in \mathbf{B}$ with the offset position. We use items with $s^x$ and $s^y$ ranging from 20cm to 40cm, and $s^z$ ranging from 10cm to 25cm. We fill items with paddings of the same density and train PCT policies on $setting\,1$ with the EV scheme and the load balancing constraint. We test our system with 50 random sequences and our method achieves $78.6\%$ space utilization with $50.9$ items packed. Our real-world packing video demo is also submitted with the supplemental material.

