# OpenReview forum: "Learning Efficient Online 3D Bin Packing on Packing Configuration Trees"
_ICLR.cc/2022/Conference — ICLR 2022 Poster_

### Official Review · Reviewer_Dhzn · 2021-11-01

**Correctness:** 3
**Technical Novelty And Significance:** 3
**Empirical Novelty And Significance:** Not applicable
**Recommendation:** 8
**Confidence:** 4

**Main Review:**

Strengths:

1. The idea of PCT, as a whole, is novel and makes a good sense. It is also well illustrated and easy to follow.
2. The motivation of the work is well explained and the pros and cons of different methods are clearly discussed.
3. The experiments are extensive and the proposed method based on PCT outperforms all competing methods significantly.

Weaknesses:

1. PCT is expanded with only 3 existing rules of heuristics, including CP, EP and EMS. Such a strategy seems to oversimplify the problem. There exist a couple of heuristics and some of them were summarised in:

    - A generalized reinforcement learning algorithm for online 3d bin-packing. AAAI 2020.

    - A hybrid grouping genetic algorithm for bin packing. Journal of Heuristics, 1996.

2. The authors claim that the proposed method is efficient. But there is neither theoretical analysis (e.g. some kind of complexity analysis) nor experimental results (e.g. run time, number of iterations needed for DRL convergence, visualisation of learning curves, etc.) for demonstrating it.

3. There is no real-world experiments, which makes the paper a bit weak especially because it attempts to demonstrate the efficiency of the PCT. A demo like the one provided with the following paper would be impressive.

     - PackerBot: Variable-Sized Product Packing with Heuristic Deep Reinforcement Learning. IROS 2021.

In addition, the paper could be improved if the following issues can be clarified:

1. In Section 3.3, what is the physical meaning of the pointer mechanism (Vinyals et al., 2015) employed for leaf node selection?

2. In Section 3.4, the authors mentioned “we fullfill PCT to a fixed length with dummy nodes”. But how can we find a fixed length for different bin configurations? If it is too small, it will not be enough for holding all leaf nodes; if it is too large, it will make the learning space excessively large and hurt the efficiency significantly.


**Summary Of The Paper:**

The paper presents PCT, a tree structure that integrates several packing heuristics for addressing the online 3D-BPP. The PCT 1) can be amenably incorporated into DRL as a term of the reward function and 2) makes the training tractable as it effectively trims the learning space defined by the low-level packing heuristics.

**Summary Of The Review:**

The idea of PCT is holistically novel and seems effective for the online 3D-BPP, which is a very challenging issue anyway. Therefore, although the paper has some issues as listed above, I am still slightly positve to it.

------------ Post rebuttal --------------

Most of my concerns have been well addressed during the rebuttal period of time. I'd appreciate very much the effort made by the authors and am happy to lift my recommendation to a full acceptance. Btw, it seems that some references are missing in the revised version, such as the two recent ones for 3D-BPP listed below:

- A Generalized Reinforcement Learning Algorithm for Online 3D Bin-Packing. AAAI 2020.
- PackerBot: Variable-Sized Product Packing with Heuristic Deep Reinforcement Learning. IROS 2021.

---

> ### Author Response · Authors · 2021-11-14
> **Response to Reviewer Dhzn**
>
> Thank you for appreciating our paper! We’d be happy to answer any questions that you may have.
>
> **- Combining PCT with more heuristics**
>
> * Thanks for the detailed suggestions. From your recommendations, we find three heuristics that can be combined with our PCT method in the discrete setting,  i.e. First Fit [1], Floor building [4], and Column building [4]. We integrate these three heuristics together to train our model, the test results are summarized as follows:
>
>
> * | Setting | Utilization   | Variance ($\times 10^{-3}$) |  Packed number |   Gap w.r.t. the best|
> | ----------------- | ----------------- | ----------------- | ----------------- | ----------------- |
> | $Setting\\,1$      | $58.5\\%$       | $4.0$ | $22.6$ | $23.0\\%$ |
> | $Setting\\,2$   |  $62.7\\%$       | $2.9$ | $24.2$ |  $27.1\\%$ |
> | $Setting\\,3$   |  $57.1\\%$       | $4.7$ | $22.0$ |  $24.6\\%$ |
>
>
> * We find that these heuristics don't suit our method well in the discrete setting. And these heuristics cannot be integrated with our method in the continuous setting since the candidate placements will be infinite. For example,  the Floor building [4] heuristic places items at the lowest positions in the bin but these positions are infinite in a continuous plane. Finding and designing better heuristics for PCT are interesting directions for future work [updated in Section 5, colored in red].
>
> **- Proofs for demonstrating PCT is efficient**
>
> * We provide our learning curves in our revised manuscript [please see Figure 4] to prove that PCT reduces the complexity for efficient learning and helps DRL methods learn better performance. The policies guided by suitable leaf node expansion schemes (e.g., EMS) outperform the policy trained on a full coordinate space during the whole training process.
>
> **- Real-world demo**
>
> * Thank you for this constructive suggestion. We are happy to provide our real-world demo at the following link. In this video, one robot arm picks items from the conveyor and packs them into the target bin with decisions given by our PCT method. Two RGB-D sensors are used for detecting the item to be packed and checking whether the packed items have deviated from their original position, respectively.
>
>
> * Real-world demo: https://drive.google.com/file/d/1MqRQDxfmrFm0pulXof4piogSGZEXTVra/view?usp=sharing
>
>
> * More details about our real-world implementation are now given in Appendix C.6.
>
> **-  The physical meaning of the pointer mechanism**
>
> * The physical meaning of the pointer mechanism is context-based attention over variable inputs.  We have added this description in our revised paper [Section 3.3, colored in red].
>
>
> * While the GAT network [3] converts the input PCT to a query vector that presents global context, we compute the compatibility of this query with all leaf nodes features with Scaled Dot-Product Attention [2] and normalize this compatibility vector with softmax operation. The final output probability distribution with a variable size equal to the length of the input leaf node set is a ‘pointer’.
>
> **- Recommendations for finding a suitable PCT length**
>
> * How to determine the suitable PCT length for different bin configurations is a good question, we give our recommendations for finding this hyperparameter and our paper is also updated accordingly [the last paragraph in Appendix A, colored in red].
>
>
> * PCT allows discarding some valid leaf nodes and this will not harm the final performance ($75.7\\%/76.0\\%$ utilization with full/fractional EV nodes).  For training, we find that the performance of trained policies is more sensitive to the number of allowed orientations $|\textbf{O}|$.  Thus we set the PCT length as  $c\times|\textbf{O}|$ where $c$ can be determined by a grid search nearby $c=25$ for different bin configurations.  For our experiments, $c=25$ performs quite well. For the test, the PCT length can be set differently from the training one, we suggest searching the test length with a validation dataset from 50 to 300 in steps of 10.
>
> **- References**
>
>
> [1] Falkenauer et al. A hybrid grouping genetic algorithm for bin packing. Journal of Heuristics, 1996.
>
> [2] Vaswani et al. Attention is all you need. NeurIPS 2017.
>
> [3] Velickovic et al. Graph attention networks. ICLR 2018.
>
> [4] Verma et al. A generalized reinforcement learning algorithm for online 3d bin-packing. AAAI 2020.

---

> ### Author Response · Authors · 2021-11-21
> **Updated Response to Reviewer Dhzn**
>
> Thank you for your updated response and your appreciation of our work!
>
> * We have added the missed reference [AAAI 2020] in our latest revision. We did our best to find the BibTeX of another paper [IROS 2021] but it seems that only one supplementary video is publicly available now. We will reference it as soon as the BibTeX of this paper is available!

---

### Official Review · Reviewer_vAx8 · 2021-11-02

**Correctness:** 4
**Technical Novelty And Significance:** 3
**Empirical Novelty And Significance:** 3
**Recommendation:** 6
**Confidence:** 2

**Main Review:**

- The reviewer is not quite familiar with this research area, but the proposed PCT and the corresponding learning method seem to be quite novel and sound.
- This paper is written in a clear and thoughtful way.
- The evaluations and related studies have been conducted thoroughly. The performance gain over existing recent baselines is clear. Empirical results on generalization also well demonstrate the efficacy of this work.

I have a few (maybe dumb) following questions:
- Could you explain the constraint in Eq. (1)? What does it mean when $e_{ij}=0$ and $p_i^d + s_i^d \le p_j^d + S^d$?
- What is the inference time for the proposed method with different leaf node expansion schemes? The theoretical time complexities have been introduced in the appendix, and the practical timing is a question of interest.
- How is the density property considered in Setting 3?


**Summary Of The Paper:**

This paper proposes a new method for online 3D bin packing problem (3D-BPP), with the newly designed packing configuration tree (PCT) for representing the state and graph attention networks (GAT) for learning the representations. Experimental comparisons with recent baselines and for different constraint settings are provided.

**Summary Of The Review:**

The reviewer is not actively working within the scope of this work. Still, it is a pleasure to read this paper, and it clearly describes and motivates the problem and proposes plausible and effective solutions with detailed experiments.

====
The reviewer has read all other reviews and the rebuttals and would very much thank the authors for the speedy and detailed response.

---

> ### Author Response · Authors · 2021-11-14
> **Response to Reviewer vAx8**
>
> Thank you for reviewing and appreciating our paper! Please let us address your concerns.
>
> **- The constraint in Eq.(1)**
>
> * Please let us explain the meaning of constraint in Eq.$\\,$(1) where $p_i^d + s_i^d \leq p_j^d + S^d (1-e_{ij}^d)$ with detail.
>
>
> * Eq.$\\,$(1) should be always satisfied when $e^d_{ij} = 0$ and $p_i^d + s_i^d - p_j^d \leq S^d$, $s_i^d$ is the item size and $S^d$ is the bin size. The condition $e^d_{ij} = 0$ means that item $j$ precedes item $i$ along  direction $d$ (i.e.$\\,$the coordinate $p^d_j < p^d_i$), so we can also get $e^d_{ji} = 1$ and the Eq.(1)
>  $p_j^d + s_j^d \leq p_i^d$ at the same time. The combination of the condition $p^d_j < p^d_i$ and  the constraint  $p_j^d + s_j^d \leq p_i^d$ is enough for avoiding overlaps. The constraint $p_i^d + s_i^d - p_j^d \leq S^d$ when  $e^d_{ij} = 0$  is always satisfied as long as Eq.$\\,$(2) where $0 \leq p_i^d \leq S^d - s_i^d$ is satisfied.
>
>
> **- The inference time of PCT**
>
>
> * Thank you for this suggestion. We agree that reporting the inference time of our PCT method is important for attracting the attention of both academia and industry. We have updated our manuscript accordingly and the results are summarized in Table 8 of Appendix C.2.
>
>
> * We have tested the time performance of both our method and all the baselines.  The inference time of PCT with different leaf node expansion schemes is all within $4.5 \times 10^{-2}$ seconds regarding all settings (CP is the fastest while EV is the slowest), which is comparable to most baselines.   We believe this performance can well meet the real industry needs.  Please see also a real industry  robot demo implemented with our PCT method:
> https://drive.google.com/file/d/1MqRQDxfmrFm0pulXof4piogSGZEXTVra/view?usp=sharing
>
> **- How is the density property considered**
>
> * The density $\rho$ for each item $n$ is sampled from the range of  $(0,1]$ uniformly [updated at the beginning of Section 4, colored in red].
>
>
> * The mass of item $n$ for checking stability is scaled to $\rho \cdot s^x_ns^y_ns^z_n$. Our method can utilize this additional property and achieve similar performance as the setting where $\rho$ is only set to 1 ($76.0\\%/75.7\\%$ utilization with the same/different density), while the performance of our baseline learning-based method deteriorates ($70.9\\%/59.6\\%$ utilization with the same/different density).

---

### Official Review · Reviewer_MYY8 · 2021-11-08

**Correctness:** 3
**Technical Novelty And Significance:** 3
**Empirical Novelty And Significance:** 3
**Recommendation:** 6
**Confidence:** 3

**Main Review:**

Strengths:

- structuring the state and action space in the proposed manner has many clear advantages over previous methods. This is demonstrated in quantitative comparisons and in the model's capacity to handle continuous spaces. Intuitively there are properties of a tree-based representation and the attention-based action space that would lend themselves better to learning.
- in the ablations in Table 5 we see that when introducing additional constraints the proposed method outperforms Zhao et al both in utilitzing space and meeting the additional task objective of the target constraint
- overall the paper is written clearly (a couple confusing spots here and there) but the overall organization is good and I think appropriate choices are made as to what details to leave in the appendices

Weaknesses:

There are some additional insights that would perhaps offer a clearer picture of what this model is learning to do and which parts of the proposed approach are necessary to do well in this problem setting:
- it might be nice to have some qualitative understanding of the learned leaf selection process. What behavior does the trained policy exhibit? Does the distribution over available leaves show clear preference for certain locations or is it somewhat uniform over possible options?
- The random baseline as described in Table 1 seems somewhat weak, for example, how does a random policy perform given the available leaves to choose from in the PCT & EV baseline?
- we see in Table 2 that removing the internal node set B altogether drops performance, but not too much. Are there simple alternatives to the state representation provided by processing a graph network over the nodes of B that might have performed just as well? Would there be any scaling issues when processing B with a large number of nodes?

I was confused about section 4.4, what exactly is being changed here? There is some set of items I and you are changing the probability of sampling a given object in that set? From Table 4 it doesn't seem like this is a particularly different setting that the model might have any difficulty with, the scores are the same across the board. How does the model handle items it has never seen? Or items with sizes that are far outside of the distribution of the items it has been trained on?

**Summary Of The Paper:**

This paper addresses the problem of online 3D bin packing where the order of objects is out of the model's control and it must make placement decisions one object at a time. Training is framed as a deep RL problem closely following recent work [Zhao et al]. The main contribution is a rethinking of the state and action space which yields much better performance. Prior work trained models to predict placement in a voxelized grid that led to a large action space that could not scale well. Instead, this work represents existing objects and potential placement locations as nodes in a tree which are processed with a graph network. Given a new object, placement nodes can be heuristically instantiated to cover valid locations, and the model uses attention to output a distribution over these nodes and determine the object placement.

**Summary Of The Review:**

I think the method presented in this paper is a good strategy for defining the state/action space for bin packing. The paper is clear and the experiments are convincing, so I recommend acceptance. It would be stronger with more compelling evidence that the graph network is learning non-trivial policy behavior, but overall the results are certainly strong over prior work and existing heuristics.

---

> ### Author Response · Authors · 2021-11-15
> **Response to Reviewer MYY8 (2/2)**
>
> **- Explanations of our generalization evaluation**
>
> * We change the probability of sampling each item from a finite item set. This probability disturbance is **fixed** during sampling a complete test sequence. The state transition of DRL $\mathcal{P}(s_{t+1}|s_t)$ which is partly determined by the disturbed probabilities of sampling items will be different from the training transition with a uniform distribution. Generalizing to a new transition is a classic challenge for reinforcement learning [2] and our method maintains its performance with fluctuation of less than $1.0\\%$.  Our original generalization evaluation (Section 4.4) has been moved to Appendix C.1 now with more detailed explanations.
>
> **- Redesign of generalization evaluation**
>
> * For providing more convincing results about the generalization ability of our method, we conduct more experiments following your suggestions. And we have updated our paper accordingly.
>
>
> * For testing our model on items it has never seen, we randomly remove $|\mathcal{I}|/5$ items from $\mathcal{I}$ for training in the discrete setting and test the trained policies on full $\mathcal{I}$, following [1]. For $setting\\,1$, the generalization results on the dataset (full $\mathcal{I}$) where unseen items exist are $74.6\\%$ utilization, while the policy trained on full $\mathcal{I}$ achieves $76.0\\%$ utilization. Please see Appendix C.1 for more results.
>
>
> * For testing our model on items with sizes that are far different from the training distribution, we test the policies trained on the uniform distribution $U(0.1, 0.5)$ with three normal distributions $N(0.3, 0.1^2)$, $N(0.1, 0.2^2)$, and $N(0.5, 0.2^2)$ within the size range [0.1, 0.5], in the continuous setting. The larger the mean value of the normal distribution, the larger sizes of sampled items. The generalization results on $setting\\,1$ are $66.1\\%$, $68.5\\%$, and $65.1\\%$ utilization for these three normal distributions respectively, while $65.3\\%$ on its original uniform distribution. Please see our new Section 4.4 for more results.
>
> **- References**
>
> [1] Zhao et al. Online 3D bin packing with constrained deep reinforcement learning. AAAI 2021.
>
> [2] Taylor & Stone. Transfer Learning for Reinforcement Learning Domains: A Survey. JMLR 2009.
>
> [3] Zhao et al. Learning practically feasible policies for online 3D bin packing. Science China Information Sciences 2021.

---

> ### Author Response · Authors · 2021-11-20
> **Response to Reviewer MYY8 (1/2)**
>
> Thank you for seeing the difficulty of the problem and appreciating our paper! Please let us address your concerns.
>
> **- Qualitative understanding of the learned policy**
>
> * Indeed, the qualitative understanding of learned policies is important, especially for industry concerns. We have visualized our packing sequences and given our analysis in our revised paper [Appendix C.4].
>
>
> * If there is no specific packing preference, our model tends to combine items of different heights together to form a plane for supporting future ones. It assigns little items to gaps to make room for future large ones [please see Figure 7]. For 3D-BPP with load balancing, the model will keep the maximum height in the bin as low as possible and pack items layer by layer, see also our real-world packing implementation below [our real-world implementation details are now given in Appendix C.6].
>
>
> * Real-world packing behavior: https://drive.google.com/file/d/1MqRQDxfmrFm0pulXof4piogSGZEXTVra/view?usp=sharing
>
> **- Performance of random policy in the PCT \& EV baseline**
>
> * Thank you for this helpful suggestion. We have added the results of randomly choosing an EV leaf in our paper [Table 1, colored in red].
>
>
> * |Setting|Utilization|Variance ($\times 10^{-3}$)|Packed number|Gap w.r.t. the best|
> |-|-|-|-|-|
> |$Setting\\,1$|$45.7\\%$|$13.5$|$18.4$|$39.9\\%$|
> |$Setting\\,2$|$51.0\\%$|$8.3$|$20.4$|$40.7\\%$|
> |$Setting\\,3$|$45.1\\%$|$12.5$|$18.1$|$40.4\\%$|
>
> **- Why removing internal nodes drops performance not too much**
>
> * The internal nodes **B** function more on settings where complex constraints between packed items are considered. We have added more explanations in our revision[Section 4.2, colored in red].
>
>
> * For $setting\\,1$ where the stability of packed items is considered, PCT achieves $76.0\\%/70.9\\%$ utilization with/without **B**. For $setting\\,2$ where items can be packed in an arbitrary fashion without considering constraints with internal nodes, the presence of **B** indeed function little ($85.3\\%/84.1\\%$ utilization with/without **B**).
>
> **- State representation alternatives**
> * The choice of the state representation for online BPP depends on the packing preferences.
>
>
> * If the problem does not involve mutual constraints between items, such as stability, isle friendliness, etc, we can remove **B** from PCT and this will not degrade performance too much as discussed before. If the problem is in a discrete setting with limited discretization, we can also use a heightmap representation like [3], which is a height matrix of the packed items' upper frontier, and use CNN to extract features. If constraints between packed items exist, a graph presentation performs currently the best. It achieves $76.0\\%$ utilization for graph presentation on stability setting while $70.9\\%,69.2\\%$, and $64.1\\%$ for heightmap, independent nodes, and sequential nodes representations (please see Section 4.2).
> Finding other alternatives for better representing 3D-BPP with different packing preferences is also an interesting direction for future work [updated in Section 5, colored in red].
>
> **- Scalability of PCT**
>
> * Our method can be transferred to larger-scale problems without fine-tuning.
>
>
> * We have conducted a large-scale packing stress test on $setting\\,2$ where the most PCT nodes are generated. We limit the maximum item sizes $s^d$ to $1/5$ bin sizes $S^d$ in the continuous setting for accommodating more items. We transfer the policies trained on setting $s^d \leq S^d/2$ to this new dataset. The test results are summarized as follows:
>
>
> * |Item sizes| &#124;**B**&#124;|&#124;**L**&#124;|Utilization|Time per step|&#124;**L**&#124; /&#124;**B**&#124;|
> |-|-|-|-|-|-|
> |$ s^d \le S^d/2$|$27.0$|$197.5$|$66.3\\%$|$2.4\times 10^{-2}$|$7.31$|
> |$s^d \le S^d/5$|$185.4$|$956.5$|$61.9\\%$|$3.7\times 10^{-2}$|$5.16$|
>
>
> * Our method successfully executes packing decisions at a real-time speed with controllable PCT sizes even if the item scale |**B**| is close to two hundred. As far as we know, this is the largest scale online 3D-BPP test within a single bin among existing works.  Please see Appendix C.3 in the revised paper for more results.

---

> > ### Comment · Reviewer_MYY8 · 2021-11-29
> > **post-rebuttal**
> >
> > I appreciate the responses by the authors and their detailed responses to the reviewers comments! Having read the other reviews, I maintain my positive rating.

---

### Official Review · Reviewer_65kq · 2021-11-09

**Correctness:** 2
**Technical Novelty And Significance:** 2
**Empirical Novelty And Significance:** Not applicable
**Recommendation:** 3
**Confidence:** 4

**Main Review:**

It is important to handle Large scale packing problem with continuous action space. The authors expand the tree by using methods of heuristics, which reduces the action space to limited number of leaf nodes (actions). Then reinforcement learning is incorporated to optimize packing efficiency. As a result, the method reduces the size of action space which leads to better learning results.

However, there are some suggestions to improve the paper:
1. The method reduces the action space to discrete nodes via heuristic search, while after heuristic search, the packing action space is not fully covered because the heuristics only select part of action space. In fact, It is a trend-off between complexity and optimality. In addition, the complexity of the tree search will grow exponentially when lots of items have to be packed, which threaten the scalability of the proposed method. Worse, a large number of leaf node would makes the attention network computationally heavy since it is quadratic related to the number of nodes.
2. After determine the candidate points, the original heuristics select the placement point of new item by its own heuristic rules. Here, the authors replace the heuristic candidate points selection with DRL. But they do not compare with recent heuristics to show whether the DRL is indeed superior than these heuristic rules.
3. The dataset is too naive, which only discretizes each edge to 10, while conventional methods have much more resolution. In their continuous setting, their results degrade, but there is no explanation about this. The authors claim that the method works on continuous solution space, but they do not define the continuous solution space. Clearly, it is different from the continuous action space in reinforcement learning. In fact, the method works on discrete action space.
4. I think Section 4.4 should be modified. The distribution of adding a uniform random disturbance to each point of another uniform distribution is almost uniform. This is not the proper way to evaluate generalization.

There are some specific points:
1. The author did not mention how many bins are used in the problem settings, which should be clearly state.
2. In the second paragraph of introduction, the authors said online packing ‘imposes additional constraints and difficulty’. but it is clear that offline packing is more complex than the its online variety since it includes the item selection step before the ‘online packing’ step. In the next sentence, the authors said ‘Learning-based approaches usually perform better than heuristic methods, especially under various complicated constraints’. Why Learning-based approaches perform better under various complicated constraints?
3. In equation 1, the right side use a capital S，I didn’t understand this, should this be a lowercase s?
4. In the first paragraph of page 3,  ‘The flaw of their work is the heightmap (frontier) state representation like Zhang et al. (2021) is still used, while the underlying interactions that exist in packed items are missed’. what does the ‘underlying interactions’ mean?
5. In Section 3.1, the authors use \pi(L_t|L_t,n_t) to denote policy, but the policy is \pi(a|s) in RL. Therefore, people may be confused between the state and action when reading. This notion is very misleading. The same problem exists in the following text.


**Summary Of The Paper:**

This work proposes a tree-based learning method for online 3D packing problem. Packing configuration tree nodes is constructed using heuristic-based tree expansion, which acts as the action space of deep reinforcement learning. The tree search schema is interesting, but this work still has lots of space to improve in terms of its methodology and experiments.

**Summary Of The Review:**

The paper has studies an interesting problem but it still has lots of space to improve.

---

> ### Author Response · Authors · 2021-11-13
> **Response to Reviewer 65kq (3/3)**
>
> **- Bin number**
>
> * We focus on online 3D-BPP with a single bin. Following your suggestion, we have added this statement in our revised paper [Section 3.1, colored in red].
>
> **- ‘imposes additional constraints and difficulty’**
>
> * The difficulty we are referring to here is that online packing is more difficult to **achieve the same space utilization** as offline packing since online packing cannot change the order of items. We are not talking about this difficulty from the perspective of problem complexity. We have also made a more detailed explanation in our revised paper [Section 1, colored in red].
>
> **- Why learning-based approaches perform better under various complicated constraints**
>
> * Learning-based approaches perform better since they learn sufficiently high-quality policies through a large number of trials and errors to handle the uncertainty in the online packing process. Designing heuristics under various complicated constraints needs both premeditating many situations manually and substantial domain knowledge and these heuristics usually have modest performance.
>
>
> * Our results in Table 1 also prove this.  The worst learning-based method achieves $69.4\\%$ utilization on the stability setting, which is better than the best heuristic method ($57.7\\%$ utilization).
>
>
> **- Capital $S$ in Equation 1**
>
> * The right side of Equation 1 ($p_i^d + s_i^d \leq p_j^d + S^d (1-e_{ij}^d)$) is a capital $S$ (bin size), not a lowercase $s$ (item size).
>
>
> * When item $i$ precedes item $j$ along $d$ with $e^d_{ij} = 1$ and $e^d_{ji} = 0$ (i.e.$\\,$the coordinate $p^d_i < p^d_j$), Equation 1 is transformed to $p_i^d + s_i^d \leq p_j^d$ and $p_j^d + s_j^d - p_i^d \leq S^d$ respectively. The former (combined with the condition  $p^d_i < p^d_j$) is for avoiding overlaps while the latter will always be satisfied. This is a typical non-overlapping representation of packing problems which can also refer to [4,6].
>
> **- The meaning of ‘underlying interactions’**
>
> * The underlying interactions mean constraints that exist between the packed items, e.g., force and torque constraints [2], and constraints that items of the same category should be placed closer, etc.
>
>
> * We have changed the description ‘underlying interactions’ to ‘underlying constraints’ in our revised paper [Section2, colored in red].
>
>
> **- Notions for denoting policy**
>
> * The notion $\pi(\textbf{L}_t|\textbf{L}_t,n_t)$ is our specific design to help readers better understand our PCT method, and we are encouraged that the other reviewers thought our presentation was clear and thoughtful. There is also a strict correspondence between our PCT notions and traditional RL notions in Section 3.4.
>
>
> * The right part  $(\textbf{L}_t,n_t)$ emphasizes that the previous BPP work only makes decisions based on the remaining positions $\textbf{L}_t$ and ignores the packed items $\textbf{B}_t$ (mentioned in Section 3.1). Therefore, our packing policy is defined as $\pi(\textbf{L}_t|\mathcal{T}_t, n_t)$, where $\textbf{B}_t \in \mathcal{T}_t$ is also an important condition ($76.0\\%/70.9\\%$ utilization with/without $\textbf{B}_t$ in Section 4.2). The left part emphasizes that the output action **probability distribution** is defined on $\textbf{L}_t$ and our packing action space is **finite** and only proportional to the number of leaf nodes.
>
>
> **- References**
>
> [1] Karabulut \& Inceoglu. A hybrid genetic algorithm for packing in 3d with deepest bottom left with fill method. ADVIS 2004.
>
> [2] Ramos et al. A physical packing sequence algorithm for the container loading problem with static mechanical equilibrium conditions. Int. Trans. Oper. Res. 2016.
>
> [3] Ha et al. An online packing heuristic for the three-dimensional container loading problem in dynamic environments and the physical internet. EvoApplications 2017.
>
> [4] Hu et al. Solving a new 3D bin packing problem with deep reinforcement learning method. arxiv:1708.05930.
>
> [5] Wang \& Hauser. Stable bin packing of non-convex 3D objects with a robot manipulator. ICRA 2019.
>
> [6] Gzara et al. The pallet loading problem: Three-dimensional bin packing with practical constraints. Eur. J. Oper. Res. 2020.
>
> [7] Yang et al. PackerBot: Variable-Sized Product Packing with Heuristic Deep Reinforcement Learning. IROS 2021.
>
> [8] Hu et al. Tap-net: transport-and-pack using reinforcement learning. TOG 2021.
>
> [9] Zhang et al. Attend2pack: Bin packing through deep reinforcement learning with attention. ICML workshop 2021.
>
> [10] Zhao et al. Online 3D bin packing with constrained deep reinforcement learning. AAAI 2021.
>
> [11] Zhao et al. Learning practically feasible policies for online 3D bin packing. Science China Information Sciences 2021.
>
> [12] Taylor & Stone. Transfer Learning for Reinforcement Learning Domains: A Survey. JMLR 2009.

---

> ### Author Response · Authors · 2021-11-13
> **Response to Reviewer 65kq (2/3)**
>
> **- Dataset choice**
>
> * Even if the bin sizes $S^d$ of our discrete dataset are set to 10, this problem is inherently challenging.
> It has about $2^{1000}$ different voxel combinations for DRL states.
> The different item orders and the different actions selected by the DRL agent also make this problem more complicated.
>
>
> * The reason we choose this dataset is to facilitate comparison with baselines [1,5] whose running costs are sensitive to discretization accuracy (mentioned at the beginning of Section 4). This dataset has also been widely used by other works [7,9,10,11] for online 3D-BPP research.
>
>
> * In Section 4.3, we also apply our methods to solving online 3D-BPP on the continuous setting where the item sizes, bin sizes, and output coordinates are all continuous values. Although the **infinite item set** (i.e.$\,|\mathcal{I}|= \infty$) increases the solving difficulty, our method still maintains about a $30\\%$ gap superior to all baselines on settings where stability constraints are considered.
>
> **- Definition of continuous solution space and action space**
>
> * The concept of continuous solution space is with respect to **the packing problems** and it indicates that the packing coordinates are continuous values. The concept of action space is w.r.t. **the packing algorithms** and describes the number of candidate actions. Our PCT method works on discrete action space and solves online 3D-BPP with continuous solution space. We have updated our paper accordingly [Section 3.4, colored in red].
>
> **- Reasons for result degradation in the continuous setting**
>
> * The result degradation compared to the discrete cases is not caused by the degradation of our method, but the fact that the **infinite item set** $|\mathcal{I}|= \infty$ **increases the difficulty** of solving this problem. We have added corresponding explanations in our revised paper [Section 4.3, colored in red].
>
>
> * The proof is that the performance of all baselines has also declined in continuous cases (see Table 3) and our method still maintains about a $30\\%$ gap superior to all baselines on settings where stability constraints are considered. If we scale the finite discrete dataset (i.e. $s_i^d/=S^d$) to the continuous domain for maintaining the same difficulty, our method achieves similar performance with the discrete cases. For $setting\\,1$ in the strict discrete sense, our method achieves $76.0\\%$ space utilization, while $76.3\\%$ on the scaled continuous cases.
>
> **- Generalization evaluation**
>
> * The disturbed distribution for each test sequence is **not uniform** anymore. Note that, the random disturbance added to the sample probability of each item is **fixed during sampling one complete sequence**. For each disturbed test sequence, the DRL state transition $\mathcal{P}(s_{t+1}|s_t)$ is different from the training transition where each item is sampled with equal probability. Generalizing to **a new transition** is a classic challenge **[12]** for reinforcement learning. Although our method maintains its performance under various amplitude disturbances, we eventually moved these results to Appendix C.1 with more detailed explanations colored in red.
>
> **- More results about generalization evaluation**
>
> * Following your suggestion, we have conducted more experiments to further discuss the generalization ability of our method.
>
>
> * For continuous cases, we test the policies trained on uniform distributions with **normal** item size distributions. The generalization results on $setting\\,1$ are $25.1$ packed items and $66.1\\%$ space utilization, while $24.9$ packed items and $65.3\\%$ space utilization on its original uniform distribution.
> Please see Section 4.4 in the revised paper for more results.
>
>
> * For discrete cases, we test our method with **unseen** items. Following [10], we randomly remove $|\mathcal{I}|/5$ items from $\mathcal{I}$ for training and test the trained policies on full $\mathcal{I}$. For $setting\\,1$, the generalization results are $28.9$ packed items and $74.6\\%$ space utilization, while the policy trained on full $\mathcal{I}$ packs $29.4$ items with $76.0\\%$ space utilization. Please see Appendix C.1 in the revised paper for more results.

---

> ### Author Response · Authors · 2021-11-13
> **Response to Reviewer 65kq (1/3)**
>
> Thanks for taking the time to review our work. We provide our response below.
>
> **- Motivation**
>
> * There is a misunderstanding about our motivation. We aim at packing regular continuous-sized items in a single bin with continuous coordinates and without discretizing the coordinate space at the expense of placement accuracy. Our method can handle the problem with a larger scale approaching two hundred smaller items (demonstrated in the following stress test for better addressing your concern), but packing excessive small items is not our core contribution. We also provide our real robot demo in a real logistics warehouse to demonstrate that the practical packing scale within a single bin is mostly in a reasonable range of 50-60 items [our real robot details are given in Appendix C.6].
>
>
> * Real robot demo: https://drive.google.com/file/d/1MqRQDxfmrFm0pulXof4piogSGZEXTVra/view?usp=sharing
>
>
> * Solving online 3D-BPP with continuous solution space is always an urgent need in the industry. To the best of our knowledge, we are the first ones to deploy the learning-based method on solving online 3D-BPP with continuous solution space successfully and we promise to publish all our code for domain development.
>
>
> **- PCT is not a trade-off between complexity and optimality**
>
> * Our PCT design reduces the complexity of online 3D-BPP to improve the performance of learning-based algorithms. Please note that online 3D-BPP **is not a search problem** (only the current item is available) but a problem of **predicting** the current placement based on past experience. Higher complexity does not necessarily lead to better results because it increases the learning burden of the DRL algorithm.
>
>
> * The results in Section 4.1 demonstrate that a full coordinate (FC) space which is the most complex is not friendly to DRL training in the discrete setting  ($86.0\\%/76.9\\%$ utilization with PCT/FC). In Section 4.3, the DRL agent which outputs continuous coordinates without our PCT module **cannot even converge**. Our PCT method outperforms the other SOTA learning-based alternatives [9,11]. We have added more descriptions in Section 4.1 which are colored in red.
>
> **- The scalability of PCT**
>
> * PCT will not grow exponentially as the number of packed items $|\textbf{B}|$ increases since **invalid leaf nodes will be removed** from leaf node set $\textbf{L}$ during the packing process, e.g., leaf nodes covered by packed items. We have added more descriptions in Section 3.1, colored in red.
>
>
> * For verifying this, we conduct a large-scale packing stress test on $setting\\,2$ where the most orientations are allowed and the most leaf nodes are generated. We limit the maximum of item sizes $s^d$ to $S^d/5$  in the continuous setting so that more small items can be accommodated ($S^d$ are bin sizes and direction $d \in \\{x,y,z\\}$). We transfer the best-performing policies (trained with EMS and $ s^d \le S^d/2$) to these new datasets **without any fine-tuning.** The statistical results are summarized as follows:
>
>
> * | Item sizes| &#124;$\textbf{B}$&#124;  | &#124;$\textbf{L}$&#124; | Utilization | Time per step |   &#124;$\textbf{L}$&#124;  /&#124;$\textbf{B}$&#124;|
> |-|-|-|-|-|-|
> |  $ s^d \le S^d/2$ |$27.0$| $197.5$ | $66.3\\%$ | $2.4\times 10^{-2}$ | $7.31$ |
> | $s^d \le S^d/5$|$185.4$| $956.5$ | $61.9\\%$ |  $3.7\times 10^{-2}$ | $5.16$ |
>
>
> * Our method successfully executes packing decisions at a real-time speed with controllable PCT sizes even if the item scale $|\textbf{B}|$ is close to two hundred. As far as we know, this is the **largest** scale packing test **within a single bin** among all existing online 3D-BPP works [7,9,10,11].
>
>
> * This experiment and more results have also been added to our revised paper,  please see Appendix C.3.
>
> **- Large leaf node set will not make the network computationally heavy**
>
> * The number of leaf nodes is controllable and our PCT method doesn't need the full leaf node set $\textbf{L}$ for inference. It randomly intercepts a subset $\textbf{L}_{sub}$ from $\textbf{L}$ if $|\textbf{L}|$ exceeds a certain length (maximum 300) for saving computing resources (mentioned in Section 3.4 and Section 4.1). This operation does not harm the final performance ($76.0\\%/75.7\\%$ utilization with/without an interception).
>
> **- Comparisons with recent heuristics**
>
> * There are extensive comparisons with recent heuristics in Table 1, including [3] (2017), [4] (2017),  [5] (2019), and [8] (2021). We also compare our method with the heuristic method proposed in [10] (2021). **[3], [4], and [10] all make decisions by choosing candidate points** (Empty Maximal Spaces)  with their own heuristic rules. Our method outperforms all baselines by a large margin (at least $24\\%$).
>
>
> *  We have marked these heuristics in red in Table 1 of our revised paper. All gathered baseline codes mentioned above will be published along with our method.

---

### Author Response · Authors · 2021-11-18
**To All Reviewers**

* We thank the reviewers for their constructive and thoughtful feedback. We are encouraged that the reviewers find our work **sound** (R2, R3, R4) and **novel** (R3, R4) with **significant better online 3D-BPP results** (R1, R2, R3, R4) and **extensive evaluations** (R2, R3, R4), the paper **clear** and **well-organized** (R2, R3, R4). They also found the approach to well handle **continuous spaces** and **additional constraints** (R2), the **generalization results well demonstrate the efficacy** (R3), and **comparisons with recent heuristics are thorough** (R2, R3, R4).


* We have revised the manuscript accordingly to resolve the reviewers’ concerns and add additional experiments and discussions to address the reviewers’ suggestions. We provide a detailed description of the changes we made in our responses below. We will further convince our readers of the practical significance of our work with our real-world demo and we promise to release all codes mentioned in our paper for domain development.


* Note: R1: Reviewer 65kq, R2: Reviewer MYY8, R3: Reviewer vAx8, R4: Reviewer Dhzn


* Real-world demo: https://drive.google.com/file/d/1MqRQDxfmrFm0pulXof4piogSGZEXTVra/view?usp=sharing

---

### Decision · Program_Chairs · 2022-01-20

**Decision:**

Accept (Poster)

**Comment:**

This paper proposes a new approach to online 3D bin packing with deep reinforcement learning. It received mixed reviews. AC finds that the responses from authors have addressed the concerns satisfactorily.